# The Step Decay Schedule: A Near Optimal, Geometrically Decaying Learning Rate Procedure For Least Squares

**Rong Ge** [1], **Sham M. Kakade** [2], **Rahul Kidambi**[3] **and Praneeth Netrapalli**[4]

[1] Duke University, [2] University of Washington, [3] Cornell University, [4] Microsoft Research, India.
rongge@cs.duke.edu, sham@cs.washington.edu, rkidambi@cornell.edu,
praneeth@microsoft.com

## Abstract

Minimax optimal convergence rates for numerous classes of stochastic convex optimization problems are well characterized, where the majority of results utilize iterate averaged stochastic gradient descent (SGD) with polynomially decaying step sizes. In contrast, the behavior of SGD's final iterate has received much less attention despite the widespread use in practice. Motivated by this observation, this work provides a detailed study of the following question: what rate is achievable using the final iterate of SGD for the streaming least squares regression problem with and without strong convexity?

First, this work shows that even if the time horizon $T$ (i.e. the number of iterations that SGD is run for) is known in advance, the behavior of SGD's final iterate with *any* polynomially decaying learning rate scheme is highly sub-optimal compared to the statistical minimax rate (by a condition number factor in the strongly convex case and a factor of $\sqrt{T}$ in the non-strongly convex case). In contrast, this paper shows that *Step Decay* schedules, which cut the learning rate by a constant factor every constant number of epochs (i.e., the learning rate decays geometrically) offer significant improvements over *any* polynomially decaying step size schedule. In particular, the behavior of the final iterate with step decay schedules is off from the statistical minimax rate by only *log* factors (in the condition number for the strongly convex case, and in $T$ in the non-strongly convex case). Finally, in stark contrast to the known horizon case, this paper shows that the anytime (i.e. the limiting) behavior of SGD's final iterate is poor (in that it queries iterates with highly sub-optimal function value infinitely often, i.e. in a limsup sense) irrespective of the stepsize scheme employed. These results demonstrate the subtlety in establishing optimal learning rate schedules (for the final iterate) for stochastic gradient procedures in fixed time horizon settings.

## 1 Introduction

Large scale machine learning relies almost exclusively on stochastic optimization methods [BB07], which include stochastic gradient descent (SGD) [RM51] and its variants [DHS11, JZ13]. In this work, we restrict our attention to the SGD algorithm where we are concerned with the behavior of the final iterate (i.e. the last point when we terminate the algorithm). A majority of (minimax optimal) theoretical results for SGD focus on polynomially decaying stepsizes [DGBSX12, RSS12, LJSB12, Bub14] (or constant stepsizes [BM13, DB15a, JKK+16] for the case of least squares regression) coupled with iterate averaging [Rup88, PJ92] to achieve minimax optimal rates of convergence. However, practical SGD implementations typically return the final iterate of a stochastic gradient procedure. This line of work in theory (based on iterate averaging) and its discrepancy with regards to

**Algorithm 1:** Step Decay scheme

**Input:** Initial vector $\mathbf{w}$, starting learning rate $\eta_0$, number of iterations $T$

**Output:** $\mathbf{w}$

for $\ell \leftarrow 1$ to $\log T$ do

$\quad \eta_\ell \leftarrow \eta_0/2^\ell$

$\quad$ for $t \leftarrow 1$ to $T/\log T$ do

$\quad\quad \mathbf{w} \leftarrow \mathbf{w} - \eta_\ell \widehat{\nabla} f(\mathbf{w})$

$\quad$ end

end

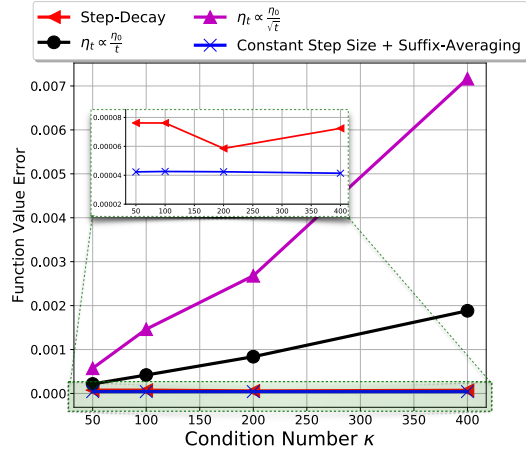

Figure 1: (Left) The Step Decay scheme for stochastic gradient descent. Note that the algorithm requires just two parameters - the starting learning rate $\eta_0$ and number of iterations $T$.
(Right) Plot of function value error vs. condition number for the final iterate of polynomially decaying stepsizes i.e., equation(5,6), step-decay schedule (Algorithm 1) compared against the minimax optimal suffix averaged iterate with a constant stepsize [JKK$^+$16] for a synthetic two-dimensional least squares regression problem(1). The condition number $\kappa$ is varied as $\{50, 100, 200, 400\}$. Exhaustive grid search is performed on starting stepsize and decay parameters. Initial excess risk is $d\sigma^2$ and the algorithm is run for $T = \kappa_{\max}^2 = 400^2$ steps (for all experiments); results are averaged over 5 random seeds. Observe that the final iterate's error grows linearly as a function of the condition number $\kappa$ for the polynomially decaying stepsize schemes, whereas, the error does not grow as a function of $\kappa$ for the geometric "step-decay" stepsize scheme. See section E.1 in supplementary material for details.

practice leads to the question with regards to the behavior of SGD's final iterate. Indeed, this question has motivated several efforts in stochastic convex optimization literature as elaborated below.

**Non-Smooth Stochastic Optimization:** The work of [Sha12] raised the question with regards to the behavior of SGD's final iterate for non-smooth stochastic optimization (with/without strong convexity). The work of [SZ12] answered this question, indicating that SGD's final iterate with polynomially decaying stepsizes achieves near minimax rates (up to $\log$ factors) in an anytime (i.e. in a limiting) sense (when number of iterations SGD is run for is not known in advance). Under specific choices of step size sequences, [SZ12]'s result on SGD's final iterate is tight owing to the recent work of [HLPR18]. More recently [JNN19] presented an approach indicating that a more nuanced stepsize sequence serves to achieve minimax rates (up to constant factors) for the non-smooth stochastic optimization setting when the end time $T$ is known in advance.

**Least Squares Regression (LSR):** In contrast to the non-smooth setting, the state of our understanding of SGD's final iterate for smooth stochastic convex optimization, or, say, the streaming least squares regression setting is far less mature − this gap motivates our paper's contributions. In particular, this paper studies SGD's final iterate behavior under various stepsize choices for least squares regression (with and without strong convexity). The use of SGD's final iterate for the least mean squares objective has featured in several efforts [WH60, Pro74, WS85, RS90], but these results *do not* achieve minimax rates of convergence, which leads to the following question:

**"** Can polynomially decaying stepsizes (known to achieve minimax rates when coupled with iterate averaging [Rup88, PJ92]) offer minimax optimal rates on SGD's *final* iterate when optimizing the streaming least squares regression objective? If not, is there *any* other family of stepsizes that can guarantee minimax rates on the final iterate of stochastic gradient descent? **"**

This paper presents progress on answering the above question − refer to contributions below for more details. Note that the oracle model employed by this work (to quantify SGD's final iterate behavior) has featured in a string of recent results that present a non-asymptotic understanding of SGD for least squares regression, with the caveat being that these results crucially rely on *iterate averaging* with constant stepsize sequences [BM13, DB15a, JKK$^+$16, JKK$^+$17b, JKK$^+$17a, NR18].

**Our contributions:** This work establishes upper and lower bounds on the behavior of SGD's final iterate, as run with standard polynomially decaying stepsizes as well as *step decay* schedules which

Table 1: Comparison of sub-optimality for *final* iterate of SGD (i.e., $\mathbb{E}\left[f(\mathbf{w}_T)\right] - f(\mathbf{w}^*)$) for stochastic convex optimization problems. This paper's focus is on SGD's final iterate for streaming least squares regression. The minimax rate refers to the best possible worst case rate with access to stochastic gradients (typically achieved with iterate averaging methods [PJ92, DGBSX12, RSS12]); the red shows the multiplicative factor increase (over the minimax rate) using the final iterate, under two different learning rate schedules - the polynomial decay and the step decay (refer to Algorithm 1). Polynomial decay schedules are of the form $\eta_t \propto 1/t^\alpha$ (for appropriate $\alpha \in [0.5, 1]$). For the general convex cases below, the final iterate with a polynomial decay scheme is off minimax rates by a $\log T$ factor (in an anytime/limiting sense) [SZ12]. Here $\widehat{\nabla}f, \nabla f = \mathbb{E}\left[\widehat{\nabla}f\right], \nabla^2 f$ denotes the stochastic gradient, gradient and the Hessian of the function $f$. With regards to least squares, we assume equation (3), following recent efforts [BM13, DB15a, JKK$^+$16]. While polynomially decaying stepsizes are nearly minimax optimal for general (strongly) convex functions, this paper indicates they are highly suboptimal on the final iterate for least squares. The geometrically decaying Step Decay schedule (Algorithm 1) provides marked improvements over any polynomial decay scheme on the final iterate for least squares. For simplicity of presentation, the results for least squares regression do not show dependence on initial error. See Theorems 1 and 2 for precise statements (and [NY83, SZ12, HLPR18] for precise statements of the general case).

| | Assumptions | Minimax rate | Rate w/ Final iterate using best poly-decay | Rate w/ Final iterate using Step Decay |
|---|---|---|---|---|
| General convex functions | $\mathbb{E}\left[\left\|\widehat{\nabla}f\right\|^2\right] \le G^2$<br>Diam(ConstraintSet) $\le D$ | $\frac{GD}{\sqrt{T}}$ | $\Theta\left(\frac{GD}{\sqrt{T}} \cdot \log T\right)$<br>[SZ12, HLPR18] | – |
| Non-strongly convex least squares regression | Eq. (3) | $\frac{\sigma^2 d}{T}$ | $\Omega\left(\frac{\sigma^2 d}{T} \cdot \frac{\sqrt{T}}{\log T}\right)$<br>(This work - Theorem 1) | $\mathcal{O}\left(\frac{\sigma^2 d}{T} \cdot \log T\right)$<br>(This work - Theorem 2) |
| General strongly convex functions | $\mathbb{E}\left[\left\|\widehat{\nabla}f\right\|^2\right] \le G^2$<br>$\nabla^2 f \succeq \mu\mathbf{I}$ | $\frac{G^2}{\mu T}$ | $\Theta\left(\frac{G^2}{\mu T} \cdot \log T\right)$<br>[SZ12, HLPR18] | – |
| Strongly convex least squares regression | Eq. (3)<br>$\nabla^2 f \succeq \mu\mathbf{I}$ | $\frac{\sigma^2 d}{T}$ | $\Omega\left(\frac{\sigma^2 d}{T} \cdot \kappa\right)$<br>(This work - Theorem 1) | $\mathcal{O}\left(\frac{\sigma^2 d}{T} \cdot \log T\right)$<br>(This work - Theorem 2) |

tends to cut the stepsize by a constant factor after every constant number of epochs (see algorithm 1), by considering the streaming least squares regression problem (with and without strong convexity). Our main result indicates that step decay schedules offer significant improvements in achieving near minimax rates over polynomially decaying stepsizes in the known horizon case (when the end time $T$ is known in advance). Figure 1 illustrates that this difference is evident (empirically) even when optimizing a two-dimensional synthetic least squares objective. Table 1 provides a summary. Finally, we present results that indicate the subtle (yet significant) differences between the known time horizon case and the anytime (i.e. the limiting) behavior of SGD's final iterate (see below). Note that proofs of our main claims can be found in the supplementary material.

Our main contributions are as follows:

- *Sub-optimality of polynomially decaying stepsizes:* For the strongly convex least squares case, this work shows that the final iterate of a polynomially decaying stepsize scheme (i.e. with $\eta_t \propto 1/t^\alpha$, with $\alpha \in [0.5, 1]$) is off the minimax rate $d\sigma^2/T$ by a factor of the *condition number* of the problem. For the non-strongly convex case of least squares, we show that *any* polynomially decaying stepsize can achieve a rate no better than $d\sigma^2/\sqrt{T}$ (up to $\log$ factors), while the minimax rate is $d\sigma^2/T$.

- *Near-optimality of the step-decay scheme:* Given a fixed end time $T$, the step-decay scheme (algorithm 1) presents a final iterate that is off the statistical minimax rate by just a $\log(T)$ factor when optimizing the strongly convex and non-strongly convex least squares regression [1], thus indicating vast improvements over polynomially decaying stepsize schedules. We note here that our Theorem 2 for the non-strongly case offers a rate on the initial error (i.e., the bias term) that is off the best known rate [BM13] (that employs iterate averaging) by a dimension factor. That said, Algorithm 1 is rather straightforward and employs the knowledge of just an initial learning rate and number of iterations for its implementation.

- *SGD has to query bad iterates infinitely often:* For the case of optimizing strongly convex least squares regression, this work shows that any stochastic gradient procedure (in a $\lim \sup$ sense) *must* query sub-optimal iterates (off by nearly a condition number) infinitely often.

- Complementary to our theoretical results for the stochastic linear regression, we evaluate the empirical performance of different learning rate schemes when training a residual network on the cifar-10 dataset and observe that the continuous variant of step decay schemes (i.e. an exponential decay) indeed compares favorably to polynomially decaying stepsizes.

While the upper bounds established in this paper (section 3.2) merit extensions towards broader smooth convex functions (with/without strong convexity), the lower bounds established in sections 3.1, 3.3 present implications towards classes of smooth stochastic convex optimization. Even in terms of upper bounds, note that there are fewer results on non-asymptotic behavior of SGD (beyond least squares) when working in the oracle model considered in this work (see below). [BM11, BM13, Bac14, NSW16] are exceptions, yet they do not achieve minimax rates on appropriate problem classes; [FGKS15] does not work in standard stochastic first order oracle model [NY83, ABRW12], so their work is not directly comparable to examine extensions towards broader function classes.

As a final note, this paper's result on the sub-optimality of standard polynomially decaying stepsizes for classes of smooth and strongly convex optimization doesn't contradict the (minimax) optimality results in stochastic approximation [PJ92]. Iterate averaging coupled with polynomially decaying learning rates (or constant learning rates for least squares [BM13, DB15a, JKK+16]) does achieve minimax rates [Rup88, PJ92]. However, as mentioned previously, this work deals with SGD's final iterate behavior (i.e. without iterate averaging), since this bears more relevance towards practice.

**Related work:** [RM51] introduced the stochastic approximation problem and Stochastic Gradient Descent (SGD). They present conditions on stepsize schemes satisfied by asymptotically convergent algorithms: these schemes are referred to as "convergent" stepsize sequences. [Rup88, PJ92] proved the asymptotic optimality of iterate averaged SGD with larger stepsize sequences. In terms of oracle models and notions of optimality, there exists two lines of thought (see also [JKK+17b]):

*Towards statistically optimal estimation procedures:* The goal of this line of thought is to match the excess risk of the statistically optimal estimator [Anb71, KC78, PJ92, LC98] on every problem instance. Several efforts consider SGD in this oracle [BM11, Bac14, DB15b, FGKS15, NSW16] presenting non-asymptotic results, often with iterate averaging. With regards to least squares, [BM13, DB15a, FGKS15, JKK+16, JKK+17b, NR18] use constant step-size SGD with iterate averaging to achieve minimax rates (on a per-problem basis) in this oracle model. SGD's final iterate behavior for least squares has featured in several efforts in the signal processing/controls literature [WH60, NN67, Pro74, WS85, RS90, SSB98], without achieving minimax rates. This paper works in this oracle model and analyzes SGD's final iterate behavior with various stepsize choices.

*Towards optimality under bounded noise assumptions:* The other line of thought presents algorithms with access to stochastic gradients satisfying bounded noise assumptions, aiming to match lower bounds provided in [NY83, RR11, ABRW12]. Asymptotic properties of "convergent" stepsize schemes have been studied in great detail [KC78, BMP90, LPW92, BB99, KY03, Lai03, Bor08]. [DGBSX12, LJSB12, RSS12, GL12, GL13a, HK14, Bub14, DFB16] use iterate averaged SGD to achieve minimax rates for various problem classes non-asymptotically. [AZ18] present an alternative approach towards minimizing the gradient norm with access to stochastic gradients. As noted, [SZ12] achieves anytime optimal rates (upto a $\log T$ factor) with the final iterate of an SGD procedure, and this is shown to be tight with the recent work of [HLPR18]. [JNN19] achieve minimax rates on the final iterate using a nuanced stepsize scheme when the number of iterations is fixed in advance.

*Geometrically Decaying Stepsize Schedules* date to [Gof77]. [DD19] employ the stepdecay schedule to prove high-probability guarantees for SGD with strongly convex objectives. In stochastic optimization, several other works, including [GL13b, HK14, AFGO19, KM19] consider doubling argument based approaches, where the epoch length is doubled everytime the stepsizes are halved. The step decay schedule is employed to yield faster rates of convergence under certain growth (and related) conditions both in convex [XLY16] and non-convex settings [YYJ18, DDC19].

**Paper organization:** Section 2 describes notation and problem setup. Section 3 presents our results on the sub-optimality of polynomial decay schemes and the near optimality of the step decay scheme. Section 3.3 presents results on the anytime behavior of SGD (i.e. the asymptotic/infinite horizon case). Section 4 presents experimental results and Section 5 presents conclusions.

## 2 Problem Setup

**Notation**: We present the setup and associated notation in this section. We represent scalars with normal font $a, b, L$ etc., vectors with boldface lowercase characters $\mathbf{a}, \mathbf{b}$ etc. and matrices with boldface uppercase characters $\mathbf{A}, \mathbf{B}$ etc. We represent positive semidefinite (PSD) ordering between two matrices using $\succeq$. The symbol $\gtrsim$ represents that the inequality holds for some universal constant.

We consider here the minimization of the following expected square loss objective:

$$\min_{\mathbf{w}} f(\mathbf{w}) \text{ where } f(\mathbf{w}) \stackrel{\text{def}}{=} \tfrac{1}{2}\mathbb{E}_{(\mathbf{x},y)\sim\mathcal{D}}[(y - \langle \mathbf{w}, \mathbf{x}\rangle)^2]. \tag{1}$$

Note that the hessian of the objective $\mathbf{H} \stackrel{\text{def}}{=} \nabla^2 f(\mathbf{w}) = \mathbb{E}\left[\mathbf{x}\mathbf{x}^\top\right]$. We are provided access to stochastic gradients obtained by sampling a new example $(\mathbf{x}_t, y_t) \sim \mathcal{D}$. These examples satisfy:

$$y = \langle \mathbf{w}^*, \mathbf{x}\rangle + \epsilon,$$

where, $\epsilon$ is the noise on the example pair $(\mathbf{x}, y) \sim \mathcal{D}$ and $\mathbf{w}^*$ is a minimizer of the objective $f(\mathbf{w})$. Given an initial iterate $\mathbf{w}_0$ and stepsize sequence $\{\eta_t\}$, our stochastic gradient update is:

$$\mathbf{w}_{t+1} \leftarrow \mathbf{w}_t - \eta_t \widehat{\nabla} f(\mathbf{w}_{t-1}); \quad \widehat{\nabla} f(\mathbf{w}_t) = -(y_t - \langle \mathbf{w}_t, \mathbf{x}_t\rangle) \cdot \mathbf{x}_t. \tag{2}$$

We assume that the noise $\epsilon = y - \langle \mathbf{w}^*, \mathbf{x}\rangle \ \forall \ (\mathbf{x}, y) \sim \mathcal{D}$ satisfies the following condition:

$$\Sigma \stackrel{\text{def}}{=} \mathbb{E}\left[\widehat{\nabla} f(\mathbf{w}^*)\widehat{\nabla} f(\mathbf{w}^*)^\top\right] = \mathbb{E}_{(\mathbf{x},y)\sim\mathcal{D}}[(y - \langle \mathbf{w}^*, \mathbf{x}\rangle)^2 \mathbf{x}\mathbf{x}^\top] \preceq \sigma^2 \mathbf{H}. \tag{3}$$

Next, assume that covariates $\mathbf{x}$ satisfy the following fourth moment inequality:

$$\mathbb{E}\left[\|\mathbf{x}\|^2 \mathbf{x}\mathbf{x}^\top\right] \preceq R^2 \ \mathbf{H} \tag{4}$$

This assumption is satisfied, say, when the norm of the covariates $\sup \|\mathbf{x}\|^2 < R^2$, but is true more generally. Finally, note that both 3 and 4 are general and are used in recent works [BM13, JKK$^+$16] that present a sharp analysis of SGD for streaming least squares problem. Next, we denote by

$$\mu \stackrel{\text{def}}{=} \lambda_{\min}(\mathbf{H}), \quad L \stackrel{\text{def}}{=} \lambda_{\max}(\mathbf{H}), \text{ and }, \kappa \stackrel{\text{def}}{=} R^2/\mu$$

the smallest eigenvalue, largest eigenvalue and condition number of $\mathbf{H}$ respectively. $\mu > 0$ in the strongly convex case but not necessarily so in the non-strongly convex case (in section 3 and beyond, the non-strongly case is referred to as the "smooth" case). Let $\mathbf{w}^* \in \arg\min_{\mathbf{w}\in\mathbb{R}^d} f(\mathbf{w})$. The excess risk of an estimator $\mathbf{w}$ is $f(\mathbf{w}) - f(\mathbf{w}^*)$. Given $t$ accesses to the stochastic gradient oracle in equation 2, any algorithm that uses these stochastic gradients and outputs $\widehat{\mathbf{w}}_t$ has sub-optimality that is lower bounded by $\frac{\sigma^2 d}{t}$. More concretely, we have that [VdV00]

$$\lim_{t\to\infty} \frac{\mathbb{E}[f(\widehat{\mathbf{w}}_t)] - f(\mathbf{w}^*)}{\sigma^2 d/t} \geq 1.$$

The rate of $(1 + o(1)) \cdot \sigma^2 d/t$ is achieved using iterate averaged SGD [Rup88, PJ92] with constant stepsizes [BM13, DB15a, JKK$^+$16]. This rate of $\sigma^2 d/t$ is called the statistical minimax rate.

## 3 Main Results

Sections 3.1, 3.2 consider the fixed time horizon setting; the former presents the significant sub-optimality of polynomially decaying stepsizes on SGD's final iterate behavior, the latter section presenting the near-optimality of SGD's final iterate. Section 3.3 presents negative results on SGD's final iterate behavior (irrespective of stepsizes employed), in the anytime (i.e. limiting) sense.

### 3.1 Suboptimality of polynomial decay schemes

This section begins by showing that there exist problem instances where polynomially decaying stepsizes considered stochastic approximation theory [RM51, PJ92] i.e., those of the form $\frac{a}{b+t^\alpha}$, for any choice of $a, b > 0$ and $\alpha \in [0.5, 1]$ are significantly suboptimal (by a factor of the condition number of the problem, or by $\sqrt{T}$ in the smooth case) compared to the statistical minimax rate [KC78].

**Theorem 1.** *Under assumptions 3, 4, there exists a class of problem instances where the following lower bounds on excess risk hold on SGD's final iterate with polynomially decaying stepsizes when given access to the oracle as written in equation 2.*

*Strongly convex case: Suppose $\mu > 0$. For any condition number $\kappa$, there exists a least squares problem instance with initial suboptimality $f(\mathbf{w}_0) - f(\mathbf{w}^*) \leq \sigma^2 d$ such that, for any $T \geq \kappa^{\frac{4}{3}}$, and for all $a, b \geq 0$ and $0.5 \leq \alpha \leq 1$, and for the learning rate scheme $\eta_t = \frac{a}{b+t^\alpha}$, we have*

$$\mathbb{E}\left[f(\mathbf{w}_T)\right] - f(\mathbf{w}^*) \geq \exp\left(-\frac{T}{\kappa \log T}\right)(f(\mathbf{w}_0) - f(\mathbf{w}^*)) + \frac{\sigma^2 d}{64} \cdot \frac{\kappa}{T}.$$

*Smooth case: For any fixed $T > 1$, there exists a least squares problem instance such that, for all $a, b \geq 0$ and $0.5 \leq \alpha \leq 1$, and for the learning rate scheme $\eta_t = \frac{a}{b+t^\alpha}$, we have*

$$\mathbb{E}\left[f(\mathbf{w}_T)\right] - f(\mathbf{w}^*) \geq \left(L \cdot \|\mathbf{w}_0 - \mathbf{w}^*\|^2 + \sigma^2 d\right) \cdot \frac{1}{\sqrt{T}\log T}.$$

For both cases (with/without strong convexity), the minimax rate is $\sigma^2 d/T$. In the strongly convex case, SGD's final iterate with polynomially decaying stepsizes pays a suboptimality factor of $\Omega(\kappa)$, whereas, in the smooth case, SGD's final iterate pays a suboptimality factor of $\Omega\left(\frac{\sqrt{T}}{\log T}\right)$.

### 3.2 Near optimality of Step Decay schemes

Given the knowledge of an end time $T$ when the algorithm is terminated, this section presents the step decay schedule (Algorithm 1), which offers significant improvements over standard polynomially decaying stepsize schemes, and obtains near minimax rates (off by only a $\log(T)$ factor).

**Theorem 2.** *Suppose we are given access to the stochastic gradient oracle 2 satisfying Assumptions 3 and 4. Running Algorithm 1 with an initial stepsize of $\eta_1 = 1/(2R^2)$ allows the algorithm to achieve the following excess risk guarantees.*

- *Strongly convex case: Suppose $\mu > 0$. We have:*

$$\mathbb{E}\left[f(\mathbf{w}_T)\right] - f(\mathbf{w}^*) \leq 2 \cdot \exp\left(-\frac{T}{2\kappa \log T \log \kappa}\right)(f(\mathbf{w}_0) - f(\mathbf{w}^*)) + 4\sigma^2 d \cdot \frac{\log T}{T}.$$

- *Smooth case: We have:*

$$\mathbb{E}\left[f(\mathbf{w}_T)\right] - f(\mathbf{w}^*) \leq 2 \cdot \left(R^2 d \cdot \|\mathbf{w}_0 - \mathbf{w}^*\|^2 + 2\sigma^2 d\right) \cdot \frac{\log T}{T}$$

While theorem 2 presents significant improvements over polynomial decay schemes, as mentioned in the contributions, the above result presents a worse rate on the initial error (by a dimension factor) in the smooth case (i.e. non-strongly convex case), compared to the best known result [BM13], which relies heavily on iterate averaging to remove this factor. It is an open question with regards to whether this factor can actually be improved or not. Furthermore, comparing the initial error dependence between the lower bound for the smooth case (Theorem 1) with the upper bound for the step decay scheme, we believe that the dependence on the smoothness $L$ should be improved to one on the $R^2$.

In terms of the variance, however, note that the polynomial decay schemes, are plagued by a polynomial dependence on the condition number $\kappa$ (for the strongly convex case), and are off the minimax rate by a $\sqrt{T}$ factor (for the smooth case). The step decay schedule, on the other hand, is off the minimax rate [Rup88, PJ92, VdV00] by only a $\log(T)$ factor. It is worth noting that Algorithm 1 admits an efficient implementation in that it requires the knowledge only of $R^2$ (similar to iterate averaging results [BM13, JKK$^+$16]) and the end time $T$. Finally, note that this $\log T$ factor can be improved to a $\log \kappa$ factor for the strongly convex case by using an additional polynomial decay scheme before switching to the Step Decay scheme.

**Proposition 3.** *Suppose we are given access to the stochastic gradient oracle 2 satisfying Assumptions 3 and 4. Let $\mu > 0$ and let $\kappa \geq 2$. For any problem and fixed time horizon $T/\log T > 5\kappa$, there exists a learning rate scheme that achieves*

$$\mathbb{E}\left[f(\mathbf{w}_T)\right] - f(\mathbf{w}^*) \leq 2\exp(-T/(6\kappa \log \kappa)) \cdot (f(\mathbf{w}_0) - f(\mathbf{w}^*)) + 100 \log_2 \kappa \cdot \frac{\sigma^2 d}{T}.$$

In order to have improved the dependence on the variance from $\log(T)$ (in theorem 2) to $\log(\kappa)$ (in proposition 3), we require access to the strong convexity parameter $\mu = \lambda_{\min}(\mathbf{H})$ in addition to $R^2$ and knowledge of the end time $T$. This parallels results known for general strongly convex setting [RSS12, LJSB12, SZ12, Bub14, JNN19].

As a final remark, note that this section's results (on step decay schemes) assumed the knowledge of a fixed time horizon $T$. In contrast, most results SGD's averaged iterate obtain anytime (i.e., limiting/infinite horizon) guarantees. Can we hope to achieve such guarantees with the final iterate?

### 3.3 SGD queries bad points infinitely often

This section shows that obtaining near statistical minimax rates with the *final* iterate is not possible without knowledge of the time horizon $T$. More concretely, we show that irrespective of the learning rate sequence employed (be it polynomially decaying or step-decay), SGD *requires* to query a point with sub-optimality at least $\Omega(\kappa/\log\kappa) \cdot \sigma^2 d/T$ for infinitely many time steps $T$.

**Theorem 4.** *Suppose we are given access to a stochastic gradient oracle 2 satisfying Assumption 3, 4. There exists a universal constant $C > 0$, and a problem instance, such that for SGD algorithm with any $\eta_t \leq 1/2R^2$ for all $t$[2], we have*

$$\limsup_{T \to \infty} \frac{\mathbb{E}\left[f(\mathbf{w}_T)\right] - f(\mathbf{w}^*)}{(\sigma^2 d/T)} \geq C \frac{\kappa}{\log(\kappa + 1)}.$$

The bad points guaranteed to exist by Theorem 4 are not rare. We show that such points occur at least once in $\mathcal{O}\left(\frac{\kappa}{\log \kappa}\right)$ iterations. Refer to Theorem 16 in appendix D in supplementary material.

## 4 Experimental Results

We present experimental validation on the suitability of the Step-decay schedule (or more precisely, its continuous counterpart, which is the exponentially decaying schedule), and compare its with the polynomially decaying stepsize schedules. In particular, we consider the use of:

$$\eta_t = \frac{\eta_0}{1 + b \cdot t} \qquad (5) \qquad \eta_t = \frac{\eta_0}{1 + b\sqrt{t}} \qquad (6) \qquad \eta_t = \eta_0 \cdot \exp\left(-b \cdot t\right). \qquad (7)$$

Where, we perform a systematic grid search on the parameters $\eta_0$ and $b$. In the section below, we consider a real world non-convex optimization problem of training a residual network on the cifar-10 dataset, with an aim to illustrate the practical implications of the results described in the paper. Complete details of the setup are given in Appendix E in the supplementary material.

### 4.1 Non-Convex Optimization: Training a Residual Net on cifar-10

We consider training a $44-$layer deep residual network [HZRS16a] with pre-activation blocks [HZRS16b] (dubbed preresnet-44) on cifar-10 dataset. The code for implementing the network can be found here [3]. For all experiments, we use Nesterov's momentum [Nes83] implemented in pytorch [4] with a momentum of $0.9$, batchsize 128, 100 training epochs, $\ell_2$ regularization of $0.0005$.

Our experiments are based on grid searching for the best learning rate decay scheme on the parametric family of learning rate schemes described above (5),(6),(7); all grid searches are performed on a separate validation set (obtained by setting aside one-tenth of the training dataset) and with models trained on the remaining 45000 samples. For presenting the final numbers in the plots/tables, we employ the best hyperparameters from the validation stage and train it on the entire $50,000$ samples and average results run with 10 different random seeds. The parameters for grid searches and other details are presented in Appendix E. Furthermore, we always extend the grid so that the best performing grid search parameter lies in the interior of our grid search.

**How does the step decay scheme compare with the polynomially decaying stepsizes?** Figure 2 and Table 2 present a comparison of the performance of the three schemes (5)-(7). These results demonstrate that the exponential scheme convicingly outperforms the polynomial step-size schemes.

Table 2: Comparing Train Cross-Entropy and Test $0/1$ Error of various learning rate decay schemes for the classification task on cifar-10 using a $44-$layer residual net with pre-activations.

| Decay Scheme | Train Function Value | Test $0/1$ error |
|---|---|---|
| $O(1/t)$ (equation (5)) | $0.0713 \pm 0.015$ | $10.20 \pm 0.7\%$ |
| $O(1/\sqrt{t})$ (equation (6)) | $0.1119 \pm 0.036$ | $11.6 \pm 0.67\%$ |
| $\exp(-t)$ (equation (7)) | $\mathbf{0.0053 \pm 0.0015}$ | $\mathbf{7.58 \pm 0.21\%}$ |

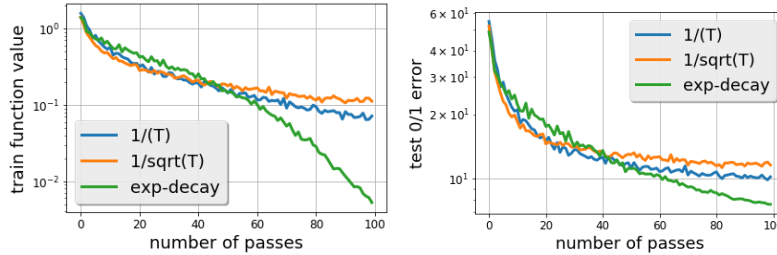

Figure 2: Plot of the training function value (left) and test $0/1-$ error (right) comparing the three decay schemes (two polynomial) 5, 6, (and one exponential) 7 scheme.

**Does suffix iterate averaging improve over final iterate's behavior for polynomially decaying stepsizes?** Towards answering this question, firstly, we consider the best performing values of equation 5 and 6, and then, average iterates of the algorithm starting from $5, 10, 20, 40, 80, 85, 90, 95, 99$ epochs when training the model for a total of $100$ epochs. While such iterate averaging (and their suffix) variants have strong theoretical support for (stochastic) convex optimization [Rup88, PJ92, RSS12, Bub14, JKK$^+$16], their impact on non-convex optimization is largely debatable. Nevertheless, this experiments's results (figure 3) indicates that suffix averaging tends to hurt the algorithm's generalization behavior (which is unsurprising given the non-convex nature of the objective). Note that, figure 3 serves to indicate that averaging the final few ($\leq 5$) epochs tends to offer nearly the same result as the final iterate's behavior, indicating that the gains of using suffix iterate averaging are relatively limited for several such settings.

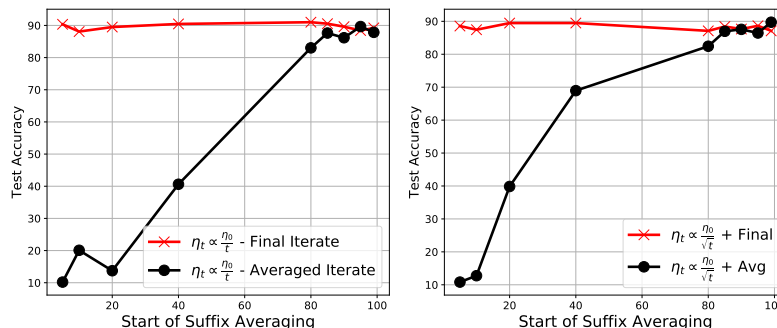

Figure 3: Performance of the suffix averaged iterate compared to the final iterate when varying the iteration when iterate averaging is begun from $\{5, 10, 20, 40, 80, 85, 90, 95, 99\}$ epochs for the $1/T$ learning rate 5 (left) and the $1/\sqrt{T}$ learning rate 6 (right).

**Does our result on "knowing" the time horizon (for step-decay schedule) present implications towards hyper-parameter search methods that work based on results extracted from truncated runs?** Towards answering this question, consider the figure 4 and Tables 3 and 4, which present a comparison of the performance of three exponential decay schemes each of which is tuned to achieve the best performance at 33, 66 and 100 epochs respectively. The key point to note is that best performing hyperparameters at 33 and 66 epochs are not the best performing at 100 epochs (which is made stark from the perspective of the validation error - refer to table 4). This demonstrates that hyper parameter selection methods that tend to discard hyper-parameters which don't perform well at

earlier stages of the optimization (i.e. based on comparing results on truncated runs), which, for e.g., is indeed the case with hyperband [LJD+17], will benefit from a round of rethinking.

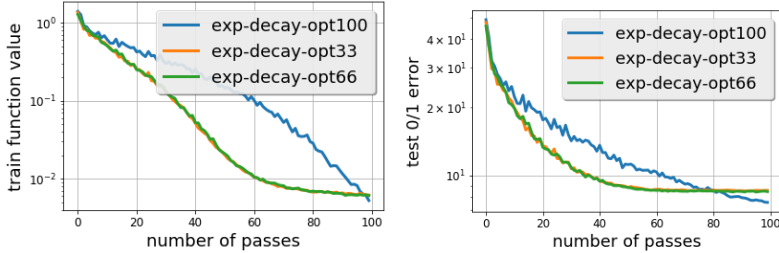

Figure 4: Plot of the training function value (left) and test $0/1-$ error (right) comparing exponential decay scheme (equation 7), with parameters optimized for 33, 66 and 100 epochs.

Table 3: Comparing training (softmax) function value by optimizing the exponential decay scheme with end times of $33/66/100$ epochs on cifar-10 dataset using a $44-$layer residual net.

| Decay Scheme | Train FVal @33 | Train FVal @66 | Train FVal @100 |
|---|---|---|---|
| $\exp(-t)$ [optimized for 33 epochs] (eqn (7)) | $\mathbf{0.098 \pm 0.006}$ | $\mathbf{0.0086 \pm 0.002}$ | $\mathbf{0.0062 \pm 0.0015}$ |
| $\exp(-t)$ [optimized for 66 epochs] (eqn (7)) | $\mathbf{0.107 \pm 0.012}$ | $\mathbf{0.0088 \pm 0.0014}$ | $\mathbf{0.0061 \pm 0.0011}$ |
| $\exp(-t)$ [optimized for 100 epochs] (eqn (7)) | $0.3 \pm 0.06$ | $0.071 \pm 0.017$ | $\mathbf{0.0053 \pm 0.0016}$ |

Table 4: Comparing test $0/1$ error by optimizing the exponential decay scheme with end times of $33/66/100$ epochs for the classification task on cifar-10 dataset using a $44-$layer residual net.

| Decay Scheme | Test 0/1 @33 | Test 0/1 @66 | Test 0/1 @100 |
|---|---|---|---|
| $\exp(-t)$ [optimized for 33 epochs] (eqn (7)) | $\mathbf{10.36 \pm 0.235}\%$ | $\mathbf{8.6 \pm 0.26}\%$ | $8.57 \pm 0.25\%$ |
| $\exp(-t)$ [optimized for 66 epochs] (eqn (7)) | $\mathbf{10.51 \pm 0.45}\%$ | $\mathbf{8.51 \pm 0.13}\%$ | $8.46 \pm 0.19\%$ |
| $\exp(-t)$ [optimized for 100 epochs] (eqn (7)) | $14.42 \pm 1.47\%$ | $9.8 \pm 0.66\%$ | $\mathbf{7.58 \pm 0.21}\%$ |

## 5    Conclusions and Discussion

The main contribution of this work shows that the behavior of SGD's final iterate for least squares regression is much more nuanced than what has been indicated by prior efforts that have primarily considered non-smooth stochastic convex optimization. The results of this paper point out the striking limitations of polynomially decaying stepsizes on SGD's final iterate, as well as sheds light on the effectiveness of starkly different schemes based on a Step Decay schedule. Somewhat coincidentally, practical implementations for certain classes of stochastic optimization do return the final iterate of SGD with step decay schedule $-$ this connection does merit an understanding through future work.

**Acknowledgments**

Rong Ge acknowledges funding from NSF CCF-1704656, NSF CCF-1845171 (CAREER), Sloan Fellowship and Google Faculty Research Award. Sham Kakade acknowledges funding from the Washington Research Foundation for Innovation in Data-intensive Discovery, NSF Award 1740551, and ONR award N00014-18-1-2247. Rahul Kidambi acknowledges funding from NSF Award 1740822.

## Footnotes

[1]This dependence can be improved to $\log$ of the condition number of the problem (for the strongly convex case) using a more refined stepsize decay scheme.

[2]Learning rate more than $2/R^2$ will make the algorithm diverge.

[3]https://github.com/D-X-Y/ResNeXt-DenseNet

[4]https://github.com/pytorch

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
