[Supplementary Material]

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

# A   Preliminaries

Before presenting the lemmas establishing the behavior of SGD under various learning rate schemes, we introduce some notation. We recount that the SGD update rule denoted through:

$$\mathbf{w}_t = \mathbf{w}_{t-1} - \eta_t \widehat{\nabla f}(\mathbf{w}_{t-1})$$

We then write out the expression for the stochastic gradient $\widehat{\nabla f}(\mathbf{w}_{t-1})$.

$$\widehat{\nabla f}(\mathbf{w}_{t-1}) = \mathbf{x}_t \mathbf{x}_t^\top (\mathbf{w}_{t-1} - \mathbf{w}^*) - \epsilon_t \mathbf{x}_t,$$

where, given the stochastic gradient corresponding to an example $(\mathbf{x}_t, y_t) \sim \mathcal{D}$, with $y_t = \langle \mathbf{w}^*, \mathbf{x}_t \rangle + \epsilon_t$, the above stochastic gradient expression naturally follows. Now, in order to analyze the contraction properties of the SGD update rule, we require the following notation:

$$P_t = \mathbf{I} - \eta_t \mathbf{x}_t \mathbf{x}_t^\top.$$

**Lemma 5.** *[For e.g. Appendix A.2.2 from [JKK$^+$16]]* **Bias-Variance tradeoff:** *Running SGD for $T-$steps starting from $\mathbf{w}_0$ and a stepsize sequence $\{\eta_t\}_{t=1}^T$ presents a final iterate $\mathbf{w}_T$ whose excess risk is upper-bounded as:*

$$\langle \mathbf{H}, \mathbb{E}\left[(\mathbf{w}_T - \mathbf{w}^*) \otimes (\mathbf{w}_T - \mathbf{w}^*)\right]\rangle \leq 2 \cdot \Bigg( \langle \mathbf{H}, \mathbb{E}\left[P_T \cdots P_1 (\mathbf{w}_0 - \mathbf{w}^*) \otimes (\mathbf{w}_0 - \mathbf{w}^*) P_1 \cdots P_T\right]\rangle$$

$$+ \left\langle \mathbf{H}, \sum_{\tau=1}^T \eta_\tau^2 \cdot \mathbb{E}\left[P_T \cdots P_{\tau+1} n_\tau \otimes n_\tau P_{\tau+1} \cdots P_T\right] \right\rangle \Bigg),$$

*where, $P_t = \mathbf{I} - \eta_t \cdot \mathbf{x}_t \mathbf{x}_t^\top$ and $n_t = \epsilon_t x_t$. Note that $\mathbb{E}\left[n_t | \mathcal{F}_{t-1}\right] = 0$ and $\mathbb{E}\left[n_t \otimes n_t | \mathcal{F}_{t-1}\right] \preceq \sigma^2 \mathbf{H}$, where, $\mathcal{F}_{t-1}$ is the filtration formed by all samples $(\mathbf{x}_1, y_1) \cdots (\mathbf{x}_{t-1}, y_{t-1})$ until time $t$.*

*Proof.* One can view the contribution of the above two terms as ones stemming from SGD's updates, which can be written as:

$$\mathbf{w}_t = \mathbf{w}_{t-1} - \eta_t \widehat{\nabla f}(w_{t-1})$$

$$\mathbf{w}_t - \mathbf{w}^* = (\mathbf{I} - \eta_t \mathbf{x}_t \mathbf{x}_t)(\mathbf{w}_{t-1} - \mathbf{w}^*) + \eta_t n_t$$

$$\mathbf{w}_t - \mathbf{w}^* = P_t \cdots P_1 (\mathbf{w}_0 - \mathbf{w}^*) + \sum_{\tau=1}^T P_t \cdots P_{\tau+1} \eta_\tau n_\tau$$

From the above equation, the result of the lemma follows straightforwardly. Now, clearly, if the noise $\epsilon$ and the inputs $\mathbf{x}$ are indepdent of each other, and if the noise is zero mean i.e. $\mathbb{E}\left[\epsilon\right] = 0$, the above inequality holds with equality (without the factor of two). This is true more generally iff

$$\mathbb{E}\left[\epsilon \mathbf{x}^{(i)} \mathbf{x}^{(j)} \mathbf{x}^{(k)}\right] = 0.$$

For more details, refer to [DB15a].

Now, in order to bound the total error, note that the original stochastic process associated with SGD's updates can be decoupled into two (simpler) processes, one being the noiseless process (which corresponds to reducing the dependence on the initial error, and is termed "bias"), i.e., where, the recurrence evolves as:

$$\mathbf{w}_t^{\text{bias}} - \mathbf{w}^* = P_t (\mathbf{w}_{t-1}^{\text{bias}} - \mathbf{w}^*) \tag{8}$$

The second recursion corresponds to the dependence on the noise (termed as variance), wherein, the process is initiated at the solution, i.e. $\mathbf{w}_0^{\text{var}} = \mathbf{w}^*$ and is driven by the noise $n_t$. The update for this process corresponds to:

$$\mathbf{w}_t^{\text{var}} - \mathbf{w}^* = P_t (\mathbf{w}_{t-1}^{\text{var}} - \mathbf{w}^*) + \eta_t n_t, \quad \text{with} \quad \mathbf{w}_0^{\text{var}} = \mathbf{w}^* \tag{9}$$

$$= \sum_{\tau=1}^t P_t \cdots P_{\tau+1} \cdot (\eta_\tau n_\tau).$$

We represent by $B_t$ the covariance of the $t^{\text{th}}$ iterate of the bias process, i.e.,

$$B_t = \mathbb{E}\left[\left(\mathbf{w}_t^{\text{bias}} - \mathbf{w}^*\right)\left(\mathbf{w}_t^{\text{bias}} - \mathbf{w}^*\right)^\top\right]$$

$$= \mathbb{E}\left[P_t B_{t-1} P_t^\top\right] = \mathbb{E}\left[P_t \cdots P_1 B_0 P_1 \cdots P_t\right]$$

The quantity that routinely shows up when bounding SGD's convergence behavior is the covariance of the variance error, i.e. $V_t := \mathbb{E}\left[(\mathbf{w}_t^{\text{var}} - \mathbf{w}^*) \otimes \mathbf{w}_t^{\text{var}} - \mathbf{w}^*)\right]$. This implies the following (simplified) expression for $V_t$:

$$V_t = \mathbb{E}\left[(\mathbf{w}_t^{\text{var}} - \mathbf{w}^*) \otimes (\mathbf{w}_t^{\text{var}} - \mathbf{w}^*)\right]$$

$$= \mathbb{E}\left[\left(\sum_{\tau=1}^t P_t \cdots P_{\tau+1} \cdot (\eta_\tau n_\tau)\right) \otimes \left(\sum_{\tau'=1}^t P_t \cdots P_{\tau'+1} \cdot (\eta_\tau' n_\tau')\right)\right]$$

$$= \sum_{\tau,\tau'} \mathbb{E}\left[P_T \cdots P_{\tau+1}(\eta_\tau n_\tau) \otimes (\eta_{\tau'} n_{\tau'}) P_{\tau'+1} \cdots P_T\right]$$

$$= \sum_{\tau=1}^T \eta_\tau^2 \mathbb{E}\left[P_T \cdots P_{\tau+1} n_\tau \otimes n_\tau P_{\tau+1} \cdots P_T\right]$$

Firstly, note that this naturally implies that the sequence of covariances $V_\tau, \tau = 1, \cdots, T$ initialized at (say), the solution, i.e., $\mathbf{V}_0 = 0$ naturally grows to its steady state covariance, i.e.,

$$V_1 \preceq V_2 \preceq \cdots \preceq V_\infty.$$

See lemma 3 of [JKK$^+$17a] for more details. Furthermore, what naturally follows in relating $V_t$ to $V_{t-1}$ is:

$$V_t \preceq \mathbb{E}\left[P_t V_{t-1} P_t^\top\right] + \eta_t^2 \sigma^2 \mathbf{H}. \tag{10}$$

$\square$

**Lemma 6** (Lemma 5 of [JKK$^+$17a]). *Running SGD with a (constant) stepsize sequence $\eta < 1/R^2$ achieves the following steady-state covariance:*

$$V_\infty \preceq \frac{\eta \sigma^2}{1 - \eta R^2} \mathbf{I}.$$

**Lemma 7.** *Suppose $\eta = 1/2R^2$, and $V_0 = \frac{\eta\sigma^2}{1-\eta R^2}\mathbf{I} = 2\eta\sigma^2\mathbf{I}$. For any sequence of learning rates $\eta_t \leq \eta = 1/2R^2 \ \forall \ t \in \{1, \cdots, t\}$, then,*

$$V_t \preceq 2\eta\sigma^2\mathbf{I} \ \ \forall \ \ t.$$

*Proof.* We will prove the lemma using an inductive argument. The base case, i.e. $t = 0$ follows from the problem statement. Note also that for SGD, $V_0 = 0$ implying the statement naturally follows. If, say, $V_t$ satisfies the equation above, from equation 10, we have the following covariance for $V_{t+1}$:

$$V_{t+1} \preceq \mathbb{E}\left[P_t V_t P_t^\top\right] + \eta_t^2 \sigma^2 \mathbf{H}$$

$$= \mathbb{E}\left[(\mathbf{I} - \eta_t \mathbf{x}_t \mathbf{x}_t^\top) V_t (\mathbf{I} - \eta_t \mathbf{x}_t \mathbf{x}_t^\top)\right] + \eta_t^2 \sigma^2 \mathbf{H}$$

$$\preceq 2\eta\sigma^2 \mathbb{E}\left[(\mathbf{I} - \eta_t \mathbf{x}_t \mathbf{x}_t^\top)(\mathbf{I} - \eta_t \mathbf{x}_t \mathbf{x}_t^\top)\right] + \eta_t^2 \sigma^2 \mathbf{H}$$

$$\preceq 2\eta\sigma^2 \mathbf{I} - 4\eta_t \eta \sigma^2 \mathbf{H} + 2\eta_t^2 \eta \sigma^2 R^2 \mathbf{H} + \eta_t^2 \sigma^2 \mathbf{H}$$

$$\preceq 2\eta\sigma^2 \mathbf{I} - 2\eta_t \eta \sigma^2 \mathbf{H} + \eta_t^2 \sigma^2 \mathbf{H}$$

$$= 2\eta\sigma^2 \mathbf{I} + \eta_t \cdot (\eta_t - 2\eta)\sigma^2 \mathbf{H}$$

$$\preceq 2\eta\sigma^2 \mathbf{I},$$

from which the lemma follows. $\square$

**Lemma 8. (Reduction from Multiplicative noise oracle)** *Let $V_t$ be the (expected) covariance of the variance error. Then, the recursion that connects $V_{t+1}$ to $V_t$ can be expressed as:*

$$V_{t+1} \preceq (\mathbf{I} - \eta_t \mathbf{H})V_t(\mathbf{I} - \eta_t \mathbf{H}) + 2\eta_t^2 \sigma^2 \mathbf{H}$$

*Proof.* From equation 10, we already know that the evolution of the co-variance of the variance error follows:

$$
\begin{aligned}
V_{t+1} &\preceq \mathbb{E}\left[P_t V_t P_t^\top\right] + \eta_t^2 \sigma^2 \mathbf{H} \\
&\preceq \mathbb{E}\left[(\mathbf{I} - \eta_t \mathbf{H}) V_t (\mathbf{I} - \eta_t \mathbf{H})\right] + \eta_t^2 \mathbb{E}\left[\mathbf{x}_t \mathbf{x}_t^\top V_t \mathbf{x}_t \mathbf{x}_t^\top\right] + \eta_t^2 \sigma^2 \mathbf{H} \\
&\preceq (\mathbf{I} - \eta_t \mathbf{H}) V_t (\mathbf{I} - \eta_t \mathbf{H}) + \eta_t^2 \left\|V_t\right\|_2 R^2 \mathbf{H} + \eta_t^2 \sigma^2 \mathbf{H} \\
&= (\mathbf{I} - \eta_t \mathbf{H}) V_t (\mathbf{I} - \eta_t \mathbf{H}) + \eta_t^2 \cdot 2\eta \sigma^2 R^2 \mathbf{H} + \eta_t^2 \sigma^2 \mathbf{H} \\
&\preceq (\mathbf{I} - \eta_t \mathbf{H}) V_t (\mathbf{I} - \eta_t \mathbf{H}) + 2\eta_t^2 \sigma^2 \mathbf{H}.
\end{aligned}
$$

Where the steps follow from lemma 7, and owing from the fact that $\eta_t \leq \eta = 1/2R^2 \ \forall \ t$. $\qquad\square$

**Note:** Basically, one could analyze an auxiliary process driven by noise with variance off by a factor of two and convert the analysis into one involving exact (deterministic) gradients.

**Lemma 9.** *[**Bias decay - strongly convex case***] Let the minimal eigenvalue of the Hessian $\mu = \lambda_{min}(\mathbf{H}) > 0$. Consider the bias recursion as in equation 8 with the stepsize set as $\eta = 1/(2R^2)$. Then,*

$$
\mathbb{E}\left[\left\|\mathbf{w}_t^{bias} - \mathbf{w}^*\right\|_2^2\right] \leq (1 - 1/(2\kappa))\mathbb{E}\left[\left\|\mathbf{w}_{t-1}^{bias} - \mathbf{w}^*\right\|_2^2\right]
$$

*Proof.* The proof follows through straight forward computations:

$$
\begin{aligned}
\mathbb{E}\left[\left\|\mathbf{w}_t^{\text{bias}} - \mathbf{w}^*\right\|_2^2\right] &\leq \mathbb{E}\left[\left\|\mathbf{w}_{t-1}^{\text{bias}} - \mathbf{w}^*\right\|_2^2\right] - 2\eta\mathbb{E}\left[\left\|\mathbf{w}_{t-1}^{\text{bias}} - \mathbf{w}^*\right\|_{\mathbf{H}}^2\right] + \eta^2 R^2 \mathbb{E}\left[\left\|\mathbf{w}_{t-1}^{\text{bias}} - \mathbf{w}^*\right\|_{\mathbf{H}}^2\right] \\
&= \mathbb{E}\left[\left\|\mathbf{w}_{t-1}^{\text{bias}} - \mathbf{w}^*\right\|_2^2\right] - \eta\mathbb{E}\left[\left\|\mathbf{w}_{t-1}^{\text{bias}} - \mathbf{w}^*\right\|_{\mathbf{H}}^2\right] \\
&\leq (1 - \eta\mu)\mathbb{E}\left[\left\|\mathbf{w}_{t-1}^{\text{bias}} - \mathbf{w}^*\right\|_2^2\right],
\end{aligned}
$$

where, the first line follows from the fact that $\mathbb{E}\left[\left\|\mathbf{x}_t\right\|_2^2 \mathbf{x}_t \mathbf{x}_t^\top\right] \preceq R^2 \mathbf{H}$ and the result follows through the definition of $\kappa$. $\qquad\square$

**Lemma 10.** *[**Reduction of the bias recursion with multiplicative noise to one resembling the variance recursion***] Consider the bias recursion that evolves as*

$$
B_t = \mathbb{E}\left[(\mathbf{w}_t - \mathbf{w}^*)(\mathbf{w}_t - \mathbf{w}^*)^\top\right] = \mathbb{E}\left[(\mathbf{I} - \gamma_t \mathbf{x}_t \mathbf{x}_t^\top) B_{t-1} (\mathbf{I} - \gamma_t \mathbf{x}_t \mathbf{x}_t^\top)\right] \quad \text{with } B_0 = (\mathbf{w}_0 - \mathbf{w}^*)(\mathbf{w}_0 - \mathbf{w}^*)^\top.
$$

*Then, the following recursion holds $\forall \gamma_t \leq 1/R^2$:*

$$
B_t \preceq (\mathbf{I} - \gamma_t \mathbf{H}) B_{t-1} (\mathbf{I} - \gamma_t \mathbf{H}) + \gamma_t^2 R^2 \left\|\mathbf{w}_0 - \mathbf{w}^*\right\|^2 \mathbf{H}.
$$

*Proof.* The result follows owing to the following computations:

$$
\begin{aligned}
B_t &= \mathbb{E}\left[(\mathbf{w}_t - \mathbf{w}^*)(\mathbf{w}_t - \mathbf{w}^*)^\top\right] \\
&= \mathbb{E}\left[(\mathbf{I} - \gamma_t \mathbf{x}_t \mathbf{x}_t^\top) B_{t-1} (\mathbf{I} - \gamma_t \mathbf{x}_t \mathbf{x}_t^\top)\right] \\
&\preceq (\mathbf{I} - \gamma_t \mathbf{H}) B_{t-1} (\mathbf{I} - \gamma_t \mathbf{H}) + \gamma_t^2 \mathbb{E}\left[(\mathbf{x}_t^\top B_{t-1} \mathbf{x}_t) \mathbf{x}_t \mathbf{x}_t^\top\right] \\
&\preceq (\mathbf{I} - \gamma_t \mathbf{H}) B_{t-1} (\mathbf{I} - \gamma_t \mathbf{H}) + \gamma_t^2 \mathbb{E}\left[\|B_{t-1}\|_2\right] R^2 \mathbf{H} \\
&\preceq (\mathbf{I} - \gamma_t \mathbf{H}) B_{t-1} (\mathbf{I} - \gamma_t \mathbf{H}) + \gamma_t^2 \mathbb{E}\left[\left\|\mathbf{w}_{t-1} - \mathbf{w}^*\right\|_2^2\right] R^2 \mathbf{H} \\
&\preceq (\mathbf{I} - \gamma_t \mathbf{H}) B_{t-1} (\mathbf{I} - \gamma_t \mathbf{H}) + \gamma_t^2 \mathbb{E}\left[\left\|\mathbf{w}_0 - \mathbf{w}^*\right\|_2^2\right] R^2 \mathbf{H},
\end{aligned}
$$

with the last inequality holding true if the squared distance to the optimum doesn't grow as a part of the recursion. We prove that this indeed is the case below:

$$
\begin{aligned}
\mathbb{E}\left[\left\|\mathbf{w}_{t-1} - \mathbf{w}^*\right\|_2^2\right] &= \mathbb{E}\left[\left\|\mathbf{w}_{t-2} - \gamma_{t-1} \mathbf{x}_{t-1} \mathbf{x}_{t-1}^\top - \mathbf{w}^*\right\|_2^2\right] \\
&\leq \mathbb{E}\left[\left\|\mathbf{w}_{t-2} - \mathbf{w}^*\right\|_2^2\right] - 2\gamma_{t-1} \mathbb{E}\left[\left\|\mathbf{w}_{t-2} - \mathbf{w}^*\right\|_{\mathbf{H}}^2\right] + \gamma_{t-1}^2 R^2 \mathbb{E}\left[\left\|\mathbf{w}_{t-2} - \mathbf{w}^*\right\|_{\mathbf{H}}^2\right] \\
&\leq \mathbb{E}\left[\left\|\mathbf{w}_{t-2} - \mathbf{w}^*\right\|_2^2\right] - \gamma_{t-1} \mathbb{E}\left[\left\|\mathbf{w}_{t-2} - \mathbf{w}^*\right\|_{\mathbf{H}}^2\right] \\
&\leq \mathbb{E}\left[\left\|\mathbf{w}_{t-2} - \mathbf{w}^*\right\|_2^2\right].
\end{aligned}
$$

Recursively applying the above argument yields the desired result. $\qquad\square$

**Note:** This result implies that the bias error (in the smooth non-strongly convex case of the least squares regression with multiplicative noise) can be bounded by employing a similar lemma as that of the variance, where one can look at the quantity $R^2 \cdot \|\mathbf{w}_0 - \mathbf{w}^*\|_2^2$ as the analog of the variance $\sigma^2$ that drives the process.

**Lemma 11.** *[Lower bounds on the additive noise oracle imply ones for the multiplicative noise oracle] Under the assumption that the covariance of noise $\Sigma = \sigma^2 \mathbf{H}$, the following statement holds. Let $V_t$ be the (expected) covariance of the variance error. Then, the recursion that connects $V_{t+1}$ to $V_t$ can be expressed as:*

$$V_{t+1} = \mathbb{E}\left[(\mathbf{I} - \eta_t \mathbf{x}_t \mathbf{x}_t^\top) V_t (\mathbf{I} - \eta_t \mathbf{x}_t \mathbf{x}_t^\top)\right] + \eta_t^2 \sigma^2 \mathbf{H}$$

*Then,*

$$V_{t+1} \succeq (\mathbf{I} - \eta_t \mathbf{H}) V_t (\mathbf{I} - \eta_t \mathbf{H}) + \eta_t^2 \sigma^2 \mathbf{H}$$

*Proof.* Let us consider firstly, the setting of (bounded) additive noise. Here, we have:

$$\hat{\nabla} f(\mathbf{w}_t) = \mathbf{H}(\mathbf{w}_t - \mathbf{w}^*) + \zeta_t, \text{ with } \mathbb{E}\left[\zeta_t | \mathbf{w}_t\right] = 0, \text{ and } \mathbb{E}\left[\zeta_t \zeta_t^\top | \mathbf{w}_t\right] = \sigma^2 \mathbf{H}.$$

Then, updates leading upto time $t + 1$ can be written as:

$$\mathbf{w}_{t+1} - \mathbf{w}^* = \prod_{\tau=1}^{t+1} (\mathbf{I} - \eta_\tau \mathbf{H})(\mathbf{w}_0 - \mathbf{w}^*) + \sum_{\tau'=1}^{t+1} \eta_{\tau'} \prod_{\tau=\tau'+1}^{t+1} (\mathbf{I} - \eta_\tau \mathbf{H}) \zeta_{\tau'}$$

This implies the covariance of the variance error is:

$$
\begin{aligned}
\tilde{V}_{t+1} &= \mathbb{E}\left[\left(\sum_{\tau'=1}^{t+1} \eta_{\tau'} \prod_{\tau=\tau'+1}^{t+1} (\mathbf{I} - \eta_\tau \mathbf{H}) \zeta_{\tau'}\right) \otimes \left(\sum_{\tau''=1}^{t+1} \eta_{\tau''} \prod_{\tau=\tau''+1}^{t+1} (\mathbf{I} - \eta_\tau \mathbf{H}) \zeta_{\tau''}\right)\right] \\
&= \sum_{\tau'=1}^{t+1} \eta_{\tau'}^2 \mathbb{E}\left[\prod_{\tau=\tau'+1}^{t+1} (\mathbf{I} - \eta_\tau \mathbf{H}) \zeta_{\tau'} \otimes \zeta_{\tau'} \prod_{\tau=t+1}^{\tau'+1} (\mathbf{I} - \eta_\tau \mathbf{H})\right] \\
&= (\mathbf{I} - \eta_{t+1} \mathbf{H}) V_t (\mathbf{I} - \eta_{t+1} \mathbf{H}) + \eta_{t+1}^2 \sigma^2 \mathbf{H}.
\end{aligned}
$$

Now, let us consider the statement of the lemma:

$$
\begin{aligned}
V_{t+1} &= \mathbb{E}\left[(\mathbf{I} - \eta_{t+1} \mathbf{x}_{t+1} \mathbf{x}_{t+1}^\top) V_t (\mathbf{I} - \eta_{t+1} \mathbf{x}_{t+1} \mathbf{x}_{t+1}^\top)\right] + \eta_{t+1}^2 \sigma^2 \mathbf{H} \\
&= (\mathbf{I} - \eta_{t+1} \mathbf{H}) V_t (\mathbf{I} - \eta_{t+1} \mathbf{H}) + \eta_{t+1}^2 \mathbb{E}\left[(\mathbf{x}_{t+1} \mathbf{x}_{t+1}^\top - \mathbf{H}) V_t (\mathbf{x}_{t+1} \mathbf{x}_{t+1}^\top - \mathbf{H})\right] + \eta_{t+1}^2 \sigma^2 \mathbf{H} \\
&\succeq (\mathbf{I} - \eta_{t+1} \mathbf{H}) V_t (\mathbf{I} - \eta_{t+1} \mathbf{H}) + \eta_t^2 \sigma^2 \mathbf{H}.
\end{aligned}
$$

Unrolling the above argument and straightforward induction, we see that $V_{t+1} \succeq \tilde{V}_{t+1}$, implying that the process driven by the multiplicative noise oracle can be lower bounded (in a PSD sense) by one that employs deterministic gradients with additive noise. $\qquad \square$

# B   Proofs of results in Section 3.1

**Theorem 12.** *Consider the additive noise oracle setting, where, we have access to stochastic gradients satisfying:*

$$\widehat{\nabla f}(\mathbf{w}) = \nabla f(\mathbf{w}) + \zeta = \mathbf{H}(\mathbf{w} - \mathbf{w}^*) + \zeta,$$

*where,*

$$\mathbb{E}\left[\zeta | \mathbf{w}\right] = 0, \text{ and, } \mathbb{E}\left[\zeta \zeta^\top | \mathbf{w}\right] = \sigma^2 \mathbf{H}$$

*The following lower bounds hold on the final iterate of a Stochastic Gradient procedure with access to the above stochastic gradients when using polynomially decaying stepsizes.*

***Strongly convex case***: *Suppose $\mu > 0$. For any condition number $\kappa$, there exists a problem instance with initial suboptimality $f(\mathbf{w}_0) - f(\mathbf{w}^*) \leq \sigma^2 d$ such that, for any $T \geq \kappa^{\frac{4}{3}}$, and for all $a, b \geq 0$ and $0.5 \leq \alpha \leq 1$, and for the learning rate scheme $\eta_t = \frac{a}{b+t^\alpha}$, we have*

$$\mathbb{E}\left[f(\mathbf{w}_T)\right] - f(\mathbf{w}^*) \geq \exp\left(-\frac{T}{\kappa \log T}\right)(f(\mathbf{w}_0) - f(\mathbf{w}^*)) + \frac{\sigma^2 d}{64} \cdot \frac{\kappa}{T}.$$

***Smooth case***: *For any fixed $T > 1$, there exists a problem instance such that, for all $a, b \geq 0$ and $0.5 \leq \alpha \leq 1$, and for the learning rate scheme $\eta_t = \frac{a}{b+t^\alpha}$, we have*

$$\mathbb{E}\left[f(\mathbf{w}_T)\right] - f(\mathbf{w}^*) \geq \left(L \cdot \|\mathbf{w}_0 - \mathbf{w}^*\|^2 + \sigma^2 d\right) \cdot \frac{1}{\sqrt{T}\log T}.$$

*Proof.* **Strongly convex case**: The problem instance is simple. Consider the case where the inputs are such that in every example $\mathbf{x}$, there is only one co-ordinate that is non-zero. Furthermore, let each co-ordinate be Gaussian with mean zero and variance for the first $d/2$ co-ordinates be $d\kappa/3$ whereas

the rest be 1. This implies $\mathbf{H} = \begin{bmatrix} d\kappa/3 & & & \\ & \ddots & & \\ & & 1 & \\ & & & \ddots \end{bmatrix}$, where the first $\frac{d}{2}$ diagonal entries are equal

to $\kappa/3$ and the remaining $\frac{d}{2}$ diagonal entries are equal to 1 and all the off diagonal entries are equal to zero. Furthermore, consider the noise to be additive (and independent of $\mathbf{x}$) with mean zero. Finally, let us denote by $v_t^{(i)} \overset{\text{def}}{=} \mathbb{E}\left[\left(\mathbf{w}_t^{(i)} - (\mathbf{w}^*)^{(i)}\right)^2\right]$ the variance in the $i^{\text{th}}$ direction at time step $t$. Let the initialization be such that $v_0^{(i)} = 3\sigma^2/\kappa$ for $i = 1, 2, ..., d/2$ and $v_0^{(i)} = \sigma^2$ for $i = d/2 + 1, ..., d$. This means that the variances for all directions with eigenvalue $\kappa$ remain equal as $t$ progresses and similarly for all directions with eigenvalue 1. We have

$$v_T^{(1)} \overset{\text{def}}{=} \mathbb{E}\left[\left(\mathbf{w}_T^{(1)} - (\mathbf{w}^*)^{(1)}\right)^2\right] = \prod_{j=1}^{T}(1 - \eta_j\kappa/3)^2\, v_0^{(1)} + \kappa\sigma^2/3\sum_{j=1}^{T}\eta_j^2\prod_{i=j+1}^{T}(1 - \eta_i\kappa/3)^2 \text{ and}$$

$$v_T^{(d)} \overset{\text{def}}{=} \mathbb{E}\left[\left(\mathbf{w}_T^{(d)} - (\mathbf{w}^*)^{(d)}\right)^2\right] = \prod_{j=1}^{T}(1 - \eta_j)^2\, v_0^{(d)} + \sigma^2\sum_{j=1}^{T}\eta_j^2\prod_{i=j+1}^{T}(1 - \eta_i)^2.$$

We consider a recursion for $v_t^{(i)}$ with eigenvalue $\lambda_i$ ($\kappa$ or 1). By the design of the algorithm, we know

$$v_{t+1}^{(i)} = (1 - \eta_t\lambda_i)^2 v_t^{(i)} + \lambda_i\sigma^2\eta_t^2.$$

Let $s(\eta, \lambda) = \frac{\lambda\sigma^2\eta^2}{1-(1-\eta\lambda)^2}$ be the solution to the stationary point equation $x = (1 - \eta\lambda)^2 + \lambda\sigma^2\eta^2$. Intuitively if we keep using the same learning rate $\eta$, then $v_t^{(i)}$ is going to converge to $s(\eta, \lambda_i)$. Also note that $s(\eta, \lambda) \approx \sigma^2\eta/2$ when $\eta\lambda \ll 1$.

We first prove the following claim showing that eventually the variance in direction $i$ is going to be at least $s(\eta_T, \lambda_i)$.

**Claim 1.** *Suppose $s(\eta_t, \lambda_i) \leq v_0^{(i)}$, then $v_t^{(i)} \geq s(\eta_t, \lambda_i)$.*

*Proof.* We can rewrite the recursion as

$$v_{t+1}^{(i)} - s(\eta_t, \lambda_i) = (1 - \eta_t\lambda_i)^2(v_t^{(i)} - s(\eta_t, \lambda_i)).$$

In this form, it is easy to see that the iteration is a contraction towards $s(\eta_t, \lambda_i)$. Further, $v_{t+1}^{(i)} - s(\eta_t, \lambda_i)$ and $v_t^{(i)} - s(\eta_t, \lambda_i)$ have the same sign. In particular, let $t_0$ be the first time such that $s(\eta_t, \lambda_i) \leq v_0^{(i)}$ (note that $\eta_t$ is monotone and so is $s(\eta_t, \lambda_i)$), it is easy to see that $v_t^{(i)} \geq v_0^{(i)}$ when $t \leq t_0$. Therefore we know $v_{t_0}^{(i)} \geq s(\eta_{t_0}, \lambda_i)$, by the recursion this implies $v_{t_0+1}^{(i)} \geq s(\eta_{t_0}, \lambda_i) \geq s(\eta_{t_0+1}, \lambda_i)$. The claim then follows from a simple induction. $\square$

If $s(\eta_T, \lambda_i) \geq v_0^{(i)}$ for $i = 1$ or $i = d$ then the error is at least $\sigma^2 d/2 \geq \kappa \sigma^2 d/T$ and we are done. Therefore we must have $s(\eta_T, \kappa) \leq v_0^{(1)} = 3\sigma^2/\kappa$, and by Claim 1 we know $v_T^{(1)} \geq s(\eta_T, \kappa) \geq \sigma^2 \eta_T/2$. The function value is at least

$$\mathbb{E}\left[f(\mathbf{w}_T)\right] - f(\mathbf{w}^*) \geq \frac{d}{2} \cdot \kappa \cdot v_T^{(1)} \geq \frac{d\kappa\sigma^2\eta_T}{12}.$$

To make sure $\mathbb{E}\left[f(\mathbf{w}_T)\right] - f(\mathbf{w}^*) \leq \frac{d\kappa\sigma^2}{64T}$ we must have $\eta_T \leq \frac{1}{6T}$. Next we will show that when this happens, $v_T^{(d)}$ must be large so the function value is still large.

We will consider two cases, in the first case, $b \geq T^\alpha$. Since $\frac{1}{16T} \geq \eta_T = \frac{a}{b+T^\alpha} \geq \frac{a}{2b}$, we have $\frac{a}{b} \leq \frac{1}{8T}$. Therefore $v_T^{(d)} \geq (1 - \frac{a}{b})^{2T} v_0^{(d)} \geq \sigma^2/2$, so the function value is at least $\mathbb{E}\left[f(\mathbf{w}_t)\right] \geq \frac{d}{2} \cdot v_T^{(d)} \geq \frac{d\sigma^2}{4} \geq \frac{\kappa d\sigma^2}{T}$, and we are done.

In the second case, $b < T^\alpha$. Since $\frac{1}{16T} \geq \eta_T = \frac{a}{b+T^\alpha} \geq \frac{a}{2T^\alpha}$, we have $a \leq \frac{1}{8}T^{\alpha-1}$. The sum of learning rates satisfy

$$\sum_{i=1}^T \eta_i \leq \sum_{i=1}^T \frac{a}{i^\alpha} \leq \sum_{i=1}^T \frac{1}{8}i^{-1} \leq 0.125\log T.$$

Here the second inequality uses the fact that $T^{\alpha-1}i^{-\alpha} \leq i^{-1}$ when $i \leq T$. Similarly, we also know $\sum_{i=1}^T \eta_i^2 \leq \sum_{i=1}^T (0.125)^2 i^{-2} \leq \pi^2/384$. Using the approximation $(1 - u)^2 \geq \exp(-2u - 4u^2)$ for $u < 1/4$, we get $v_T^{(d)} \geq \exp(-2\sum_{i=1}^T \eta_i - 4\sum_{i=1}^T \eta_i^2) v_0^{(d)} \geq \sigma^2/5T^{\frac{1}{4}}$, so the function value is at least $\mathbb{E}\left[f(\mathbf{w}_t)\right] \geq \frac{d}{2} \cdot v_T^{(d)} \geq \frac{d\sigma^2}{10T^{\frac{1}{4}}} \geq \frac{\kappa d\sigma^2}{32T}$. This concludes the second case and proves the strongly convex part of the theorem.

**Smooth case**: The proof of this part is quite similar to that of the strongly convex case above but with a subtle change in the initialization. In order to make this clear, we will do the proof from scratch with out borrowing anything from the previous argument. Let $\mathbf{H} = \begin{bmatrix} 1 & & & & \\ & \ddots & & & \\ & & \frac{d}{\kappa} & & \\ & & & \ddots & \end{bmatrix}$,

where the first $\frac{d}{2}$ diagonal entries are equal to $1$ and the remaining $\frac{d}{2}$ diagonal entries are equal to $\frac{d}{\kappa}$ and all the off diagonal entries are equal to zero. We will use $\kappa = \frac{1}{\sqrt{T}}$. Let us denote by $v_t^{(i)} \overset{\text{def}}{=} \mathbb{E}\left[\left(\mathbf{w}_t^{(i)} - (\mathbf{w}^*)^{(i)}\right)^2\right]$ the variance in the $i^{\text{th}}$ direction at time step $t$. Let the initialization be such that $v_0^{(i)} = \sigma^2/\kappa$ for $i = 1, 2, ..., d/2$ and $v_0^{(i)} = \sigma^2$ for $i = d/2 + 1, ..., d$. This means that the variances for all directions with eigenvalue $\kappa$ remain equal as $t$ progresses and similarly for all directions with eigenvalue $1$. We have

$$v_T^{(1)} \overset{\text{def}}{=} \mathbb{E}\left[\left(\mathbf{w}_T^{(1)} - (\mathbf{w}^*)^{(1)}\right)^2\right] = \prod_{j=1}^T (1 - \eta_j\kappa/3)^2 v_0^{(1)} + \kappa\sigma^2/3 \sum_{j=1}^T \eta_j^2 \prod_{i=j+1}^T (1 - \eta_i\kappa/3)^2 \text{ and}$$

$$v_T^{(d)} \overset{\text{def}}{=} \mathbb{E}\left[\left(\mathbf{w}_T^{(d)} - (\mathbf{w}^*)^{(d)}\right)^2\right] = \prod_{j=1}^T (1 - \eta_j)^2 v_0^{(d)} + \sigma^2 \sum_{j=1}^T \eta_j^2 \prod_{i=j+1}^T (1 - \eta_i)^2.$$

We consider a recursion for $v_t^{(i)}$ with eigenvalue $\lambda_i$ ($1$ or $\frac{1}{\kappa}$). By the design of the algorithm, we know

$$v_{t+1}^{(i)} = (1 - \eta_t\lambda_i)^2 v_t^{(i)} + \lambda_i\sigma^2\eta_t^2.$$

Let $s(\eta, \lambda) = \frac{\lambda\sigma^2\eta^2}{1-(1-\eta\lambda)^2}$ be the solution to the stationary point equation $x = (1 - \eta\lambda)^2 + \lambda\sigma^2\eta^2$. Intuitively if we keep using the same learning rate $\eta$, then $v_t^{(i)}$ is going to converge to $s(\eta, \lambda_i)$. Also note that $s(\eta, \lambda) \approx \sigma^2\eta/2$ when $\eta\lambda \ll 1$.

If $s(\eta_T, \lambda_i) \geq v_0^{(i)}$ for $i = 1$ or $i = d$ then the error is at least $\sigma^2 d/2\kappa \geq \kappa\sigma^2 d/T$ and we are done. Therefore we must have $s(\eta_T, \kappa) \leq v_0^{(1)} = 3\sigma^2/\kappa$, and by Claim 1 we know $v_T^{(1)} \geq s(\eta_T, \kappa) \geq \sigma^2\eta_T/2$. The function value is at least

$$\mathbb{E}\left[f(\mathbf{w}_T)\right] - f(\mathbf{w}^*) \geq \frac{d}{2} \cdot v_T^{(1)} \geq \frac{d\sigma^2\eta_T}{4}.$$

To make sure $\mathbb{E}\left[f(\mathbf{w}_T)\right] - f(\mathbf{w}^*) \leq \frac{d\kappa\sigma^2}{64T\log T}$ we must have $\eta_T \leq \frac{\kappa}{16T\log T}$. Next we will show that when this happens, $v_T^{(d)}$ must be large so the function value is still large.

We will consider two cases, in the first case, $b \geq T^\alpha$. Since $\frac{\kappa}{16T\log T} \geq \eta_T = \frac{a}{b+T^\alpha} \geq \frac{a}{2b}$, we have $\frac{a}{b} \leq \frac{\kappa}{8T\log T}$. Therefore $v_T^{(d)} \geq (1 - \frac{a}{b})^{2T} v_0^{(d)} \geq \sigma^2/2$, so the function value is at least $\mathbb{E}\left[f(\mathbf{w}_t)\right] - f(\mathbf{w}^*) \geq \frac{d}{2} \cdot \frac{1}{\kappa} \cdot v_T^{(d)} \geq \frac{d\sigma^2}{4\kappa} \geq \frac{\kappa d\sigma^2}{T}$, and we are done.

In the second case, $b < T^\alpha$. Since $\frac{\kappa}{16T\log T} \geq \eta_T = \frac{a}{b+T^\alpha} \geq \frac{a}{2T^\alpha}$, we have $a \leq \frac{1}{8\log T}\kappa T^{\alpha-1}$. The sum of learning rates satisfy

$$\sum_{i=1}^{T} \eta_i \leq \sum_{i=1}^{T} \frac{a}{i^\alpha} \leq \sum_{i=1}^{T} \frac{1}{8\log T}\kappa i^{-1} \leq 0.125\kappa.$$

Here the second inequality uses the fact that $T^{\alpha-1}i^{-\alpha} \leq i^{-1}$. Similarly, we also know

$$\sum_{i=1}^{T} \eta_i^2 \leq \sum_{i=1}^{T} (0.125\kappa/\log T)^2 i^{-2} \leq \pi^2\kappa^2/384.$$

Using the approximation $(1-u)^2 \geq \exp(-2u-4u^2)$ for $u < 1/4$, we get $v_T^{(d)} \geq \exp(-2\sum_{i=1}^{T} \frac{\eta_i}{\kappa} - 4\sum_{i=1}^{T} \frac{\eta_i^2}{\kappa^2})v_0^{(d)} \geq \sigma^2/5$, so the function value is at least $\mathbb{E}\left[f(\mathbf{w}_t)\right] \geq \frac{d}{2} \cdot \frac{1}{\kappa} \cdot v_T^{(d)} \geq \frac{d\sigma^2}{10\kappa} \geq \frac{d\sigma^2}{10\sqrt{T}}$. This concludes the second case and proves the strongly convex part of the theorem. Since $\|\mathbf{H}\| \cdot \|\mathbf{w}_0 - \mathbf{w}^*\|^2 = d\sigma^2$, we have

$$\mathbb{E}\left[f(\mathbf{w}_T)\right] - f(\mathbf{w}^*) \geq \sigma^2 d \cdot \min\left(\frac{\kappa}{T\log T}, \frac{1}{10\sqrt{T}}\right) \geq \left(L \cdot \|\mathbf{w}_0 - \mathbf{w}^*\|^2 + \sigma^2 d\right) \cdot \frac{1}{\sqrt{T}\log T}.$$

This proves the theorem. $\qquad\square$

*Proof of Theorem 1.* The proof of theorem 1 follows straightforwardly when combining the result of lemma 11 and theorem 12. $\qquad\square$

## C   Proofs of results in Section 3.2

**Theorem 13.** *Consider the additive noise oracle setting, where, we have access to stochastic gradients satisfying:*

$$\widehat{\nabla f}(\mathbf{w}) = \nabla f(\mathbf{w}) + \zeta = \mathbf{H}(\mathbf{w} - \mathbf{w}^*) + \zeta,$$

*where,*

$$\mathbb{E}\left[\zeta|\mathbf{w}\right] = 0, \text{ and, } \mathbb{E}\left[\zeta\zeta^\top|\mathbf{w}\right] \preceq \hat{\sigma}^2\mathbf{H}$$

*Running Algorithm 1 with an initial stepsize of $\eta_1 = 1/R^2$, starting from the solution, i.e. $\mathbf{w}_0 = \mathbf{w}^*$ allows the algorithm to obtain the following dependence on the variance error:*

$$\mathbb{E}\left[f(\mathbf{w}_T^{var})\right] - f(\mathbf{w}^*) \leq 2\frac{d\hat{\sigma}^2\log T}{T}$$

*Proof.* The learning rate scheme is as follows. Divide the total time horizon $T$ into $\log T$ phases, each of length $\frac{T}{\log T}$. In the $\ell^{\text{th}}$ phase, the learning rate is set to be $\frac{1}{2^\ell R^2}$. The variance in the $k^{\text{th}}$ coordinate can be bounded as

$$v_T^{(k)} \leq \prod_{j=1}^T \left(1 - \eta_j \lambda^{(k)}\right)^2 v_0^{(k)} + \lambda^{(k)} \hat{\sigma}^2 \sum_{j=1}^T \eta_j^2 \prod_{i=j+1}^T \left(1 - \eta_i \lambda^{(k)}\right)^2$$

$$\leq \exp\left(-2 \sum_{j=1}^T \eta_j \lambda^{(k)}\right) v_0^{(k)}$$

$$+ \lambda^{(k)} \hat{\sigma}^2 \sum_{\ell=1}^{\log T} \frac{1}{2^{2\ell}(R^2)^2} \sum_{j=1}^{T/\log T} \left(1 - \frac{\lambda^{(k)}}{2^\ell (R^2)}\right)^{2j} \cdot \prod_{u=\ell+1}^{\log T} \left(1 - \frac{\lambda^{(k)}}{2^u R^2}\right)^{T/\log T}$$

$$\leq \exp\left(-\frac{2\lambda^{(k)}}{R^2} \cdot \frac{T}{\log T}\right) v_0^{(k)} + \lambda^{(k)} \hat{\sigma}^2 \sum_{\ell=1}^{\log T} \frac{1}{2^{2\ell}(R^2)^2} \cdot \frac{2^\ell R^2}{\lambda^{(k)}} \cdot \prod_{u=\ell+1}^{\log T} \exp\left(-\frac{\lambda^{(k)}T}{2^u R^2 \log T}\right)$$

$$\leq \exp\left(-\frac{2\lambda^{(k)}}{R^2} \cdot \frac{T}{\log T}\right) v_0^{(k)} + \sum_{\ell=1}^{\log T} \frac{\hat{\sigma}^2}{2^\ell R^2} \prod_{u=\ell+1}^{\log T} \exp\left(-\frac{\lambda^{(k)}T}{2^u R^2 \log T}\right). \tag{11}$$

Let $\ell^* \overset{\text{def}}{=} \max\left(0, \lfloor \log\left(\frac{\lambda^{(k)}}{R^2} \cdot \frac{T}{\log T}\right) \rfloor\right)$. We now split the summation in the second term in (11) into two parts and bound each of them below.

$$\sum_{\ell=1}^{\ell^*} \frac{\hat{\sigma}^2}{2^\ell R^2} \prod_{u=\ell+1}^{\log T} \exp\left(-\frac{\lambda^{(k)}T}{2^u R^2 \log T}\right) \leq \sum_{\ell=1}^{\ell^*} \frac{\hat{\sigma}^2}{2^\ell R^2} \prod_{u=\ell+1}^{\ell^*} \exp\left(-\frac{\lambda^{(k)}T}{2^u R^2 \log T}\right)$$

$$\leq \sum_{\ell=1}^{\ell^*} \frac{\hat{\sigma}^2}{2^\ell R^2} \prod_{u=\ell+1}^{\ell^*} \exp\left(-2^{\ell^*-u}\right) \leq \sum_{\ell=1}^{\ell^*} \frac{\hat{\sigma}^2}{2^\ell R^2} \exp\left(-2^{\ell^*-\ell}\right)$$

$$\leq \frac{\hat{\sigma}^2}{2^{\ell^*} R^2} \sum_{\ell=1}^{\ell^*} 2^{\ell^*-\ell} \exp\left(-2^{\ell^*-\ell}\right) \leq \frac{\hat{\sigma}^2}{2^{\ell^*} R^2} \leq \frac{\hat{\sigma}^2}{\lambda^{(k)}} \cdot \frac{\log T}{T}. \tag{12}$$

For the second part, we have

$$\sum_{\ell=\ell^*+1}^{\log T} \frac{\hat{\sigma}^2}{2^\ell R^2} \prod_{u=\ell+1}^{\log T} \exp\left(-\frac{\lambda^{(k)}T}{2^u R^2 \log T}\right) \leq \sum_{\ell=\ell^*+1}^{\log T} \frac{\hat{\sigma}^2}{2^\ell R^2} \leq \sum_{\ell=\ell^*+1}^{\log T} \frac{\hat{\sigma}^2}{2^{\ell^*} R^2} \leq \frac{\hat{\sigma}^2}{\lambda^{(k)}} \cdot \frac{\log T}{T}. \tag{13}$$

Plugging (12) and (13) into (11), we obtain

$$v_T^{(k)} \leq \exp\left(-\frac{2\lambda^{(k)}}{R^2} \cdot \frac{T}{\log T}\right) v_0^{(k)} + \frac{2\hat{\sigma}^2}{\lambda^{(k)}} \cdot \frac{\log T}{T}.$$

The function suboptimality can now be bounded as

$$\mathbb{E}\left[f(\mathbf{w}_T^{\text{var}})\right] - f(\mathbf{w}^*) = \sum_{k=1}^d \lambda^{(k)} \cdot v_T^{(k)}$$

$$\leq \sum_{k=1}^d \lambda^{(k)} \left(\exp\left(-\frac{2\lambda^{(k)}}{R^2} \cdot \frac{T}{\log T}\right) v_0^{(k)} + \frac{2\hat{\sigma}^2}{\lambda^{(k)}} \cdot \frac{\log T}{T}\right).$$

$$\mathbb{E}\left[f(\mathbf{w}_T^{\text{var}})\right] - f(\mathbf{w}^*) \leq \sum_{k=1}^d \left(\frac{L \log T}{T} v_0^{(k)} + 2\hat{\sigma}^2 \cdot \frac{\log T}{T}\right) = 2\left(\hat{\sigma}^2 d\right) \frac{\log T}{T}.$$

$\square$

*Proof of Theorem 2.* **Smooth case:** The result follows by instantiating $\hat{\sigma}^2$ in theorem 13 with $2\sigma^2$ (lemma 8) and $R^2 \|\mathbf{w}_0 - \mathbf{w}^*\|_2^2$ (lemma 10) and using the lemma 5 to obtain the result.

**Strongly convex case:** As with the smooth case, the result relies on instantiating theorem 13 with $2\sigma^2$ (lemma 8) and using lemma 9 and then appealing to lemma 5. □

**Proposition 14.** *Consider the additive noise oracle setting, where, we have access to stochastic gradients satisfying:*

$$\widehat{\nabla f}(\mathbf{w}) = \nabla f(\mathbf{w}) + \zeta = \mathbf{H}(\mathbf{w} - \mathbf{w}^*) + \zeta,$$

*where,*

$$\mathbb{E}\left[\zeta|\mathbf{w}\right] = 0, \text{ and, } \mathbb{E}\left[\zeta\zeta^\top|\mathbf{w}\right] \leq \sigma^2 \mathbf{H}$$

*There exists a stepsize scheme with which, by starting at the solution (i.e. $\mathbf{w}_0 = \mathbf{w}^*$) the algorithm obtains the following dependence on the variance error, under the assumption that $\mu > 0$ and $\kappa \geq 2$.*

$$\mathbb{E}\left[f(\mathbf{w}_T^{var})\right] - f(\mathbf{w}^*) \leq 50 \log_2 \kappa \cdot \frac{\sigma^2 d}{T}.$$

*Proof.* The learning rate scheme is as follows.

We first break $T$ into three equal sized parts. Let $A = T/3$ and $B = 2T/3$. In the first $T/3$ steps, we use a constant learning rate of $1/R^2$. Note that at the end of this phase, (since $T > \kappa$) the dependence on the initial error decays geometrically. In the second $T/3$ steps, we use a polynomial decay learning rate $\eta_{A+t} = \frac{1}{\mu(\kappa+t/2)}$. In the third $T/3$ steps, we break the steps into $\log_2(\kappa)$ equal sized phases. In the $\ell^{\text{th}}$ phase, the learning rate to be used is $\frac{5 \log_2 \kappa}{2^\ell \cdot \mu \cdot T}$. Note that the learning rate in the first phase depends on strong convexity and that in the last phase depends on smoothness (since the last phase has $\ell = \log \kappa$).

Recall the variance in the $k^{\text{th}}$ coordinate can be upper bounded by

$$v_T^{(k)} \overset{\text{def}}{=} \mathbb{E}\left[\left(\mathbf{w}_T^{(k)} - (\mathbf{w}^*)^{(1)}\right)^2\right] \leq \prod_{j=1}^T \left(1 - \eta_j \lambda^{(k)}\right)^2 v_0^{(1)} + \lambda^{(k)}\sigma^2 \sum_{j=1}^T \eta_j^2 \prod_{i=j+1}^T \left(1 - \eta_i \lambda^{(k)}\right)^2$$

$$\leq \exp\left(-2\sum_{j=1}^T \eta_j \lambda^{(k)}\right) v_0^{(1)} + \lambda^{(k)}\sigma^2 \sum_{j=1}^T \eta_j^2 \exp\left(-2\sum_{i=j+1}^T \eta_i \lambda^{(k)}\right).$$

We will show that for every $k$, we have

$$v_T^{(k)} \leq \frac{v_0^{(k)}}{T^3} + \frac{50 \log_2 \kappa}{\lambda^{(k)} T} \cdot \sigma^2., \tag{14}$$

which directly implies the theorem.

We will consider the first $T/3$ steps. The guarantee that we will prove for these iterations is: for any $t \leq A$, $v_t^{(k)} \leq (1 - \lambda^{(k)}/R^2)^{2t} v_0^{(k)} + \frac{\sigma^2}{R^2}$.

This can be proved easily by induction. Clearly this is true when $t = 0$. Suppose it is true for $t - 1$, let's consider step $t$. By recursion of $v_t^{(k)}$ we know

$$v_t^{(k)} = (1 - \lambda^{(k)}/R^2)^2 v_{t-1}^{(k)} + \lambda^{(k)}\sigma^2/(R^2)^2$$

$$\leq (1 - \lambda^{(k)}/R^2)^{2t} v_0^{(k)} + \frac{\sigma^2}{R^2}\left((1 - \lambda^{(k)}/R^2)^2 + \lambda^{(k)}/R^2\right)$$

$$\leq (1 - \lambda^{(k)}/R^2)^{2t} v_0^{(k)} + \frac{\sigma^2}{R^2}.$$

Here the second step uses induction hypothesis and the third step uses the fact that $(1-x)^2 + x \leq 1$ when $x \in [0, 1]$. In particular, since $(1-\lambda^{(k)}/R^2)^{2T/3} \leq (1-1/\kappa)^{2T/3} \leq (1-1/\kappa)^{3\kappa \log T} = 1/T^3$, we know at the end of the first phase, $v_A^{(k)} \leq v_0^{(k)}/T^3 + \frac{\sigma^2}{R^2}$.

In the second $T/3$ steps, the guarantee would be: for any $t \leq T/3$, $v_{A+t}^{(k)} \leq v_0^{(k)}/T^3 + 2\eta_{A+t}\sigma^2$.

We will again prove this by induction. The base case ($t = 0$) follows immediately from the guarantee for the first part. Suppose this is true for $A + t - 1$, let us consider $A + t$, again by recursion we know

$$
\begin{aligned}
v_{A+t}^{(k)} &= (1 - \lambda^{(k)}\eta_{A+t-1})^2 v_{A+t-1}^{(k)} + \lambda^{(k)}\sigma^2\eta_{A+t-1}^2 \\
&\leq v_0^{(k)}/T^3 + 2\eta_{A+t-1}\sigma^2\left((1-\lambda^{(k)}\eta_{A+t-1})^2 + \frac{1}{2}\lambda^{(k)}\eta_{A+t-1}\right) \\
&\leq v_0^{(k)}/T^3 + 2\eta_{A+t-1}\sigma^2(1 - \frac{1}{2}\mu\eta_{A+t-1}) \leq v_0^{(k)}/T^3 + 2\eta_{A+t}\sigma^2.
\end{aligned}
$$

Here the last line uses the fact that $2\eta_{A+t-1}(1 - \frac{1}{2}\mu\eta_{A+t-1}) \leq 2\eta_{A+t}\sigma^2$, which is easy to verify by our choice of $\eta$. Therefore, at the end of the second part, we have $v_B^{(k)} \leq v_0^{(k)}/T^3 + \frac{2\sigma^2}{\mu(\kappa+T/6)}$.

Finally we will analyze the third part. Let $\hat{T} = T/3\log_2\kappa$, we will consider the variance $v_{B+\ell\hat{T}}^{(k)}$ at the end of each phase. We will make the following claim by induction:

**Claim 2.** *Suppose* $2^\ell \cdot \mu \leq \lambda^{(k)}$, *then*

$$
v_{B+\ell\hat{T}}^{(k)} \leq v_B^{(k)}\exp(-3\ell) + 2\hat{T}\eta_\ell^2\lambda^{(k)}\sigma^2.
$$

*Proof.* We will prove this by induction. When $\ell = 0$, clearly we have $v_B^{(k)} \leq v_B^{(k)}$ so the claim is true. Suppose the claim is true for $\ell - 1$, we will consider what happens after the algorithm uses $\eta_\ell$ for $\hat{T}$ steps. By the recursion of the variance we have

$$
v_{\ell\hat{T}}^{(k)} \leq v_{(\ell-1)\hat{T}}^{(k)} \cdot \exp(-2\eta_\ell \cdot \lambda^{(k)}\hat{T}) + \hat{T}\eta_\ell^2\lambda^{(k)}\sigma^2.
$$

Since $2^\ell \cdot \mu \leq \lambda^{(k)}$, we know $\exp(-2\eta_\ell \cdot \lambda^{(k)}\hat{T}) \leq \exp(-3)$. Therefore by induction hypothesis we have

$$
v_{B+\ell\hat{T}}^{(k)} \leq v_B^{(k)}\exp(-3\ell) + \exp(-3) \cdot 2\hat{T}\eta_{\ell-1}^2\lambda^{(k)} + \hat{T}\eta_\ell^2\lambda^{(k)} \leq v_B^{(k)}\exp(-3\ell) + 2\hat{T}\eta_\ell^2\lambda^{(k)}.
$$

This finishes the induction. $\qquad\square$

By Claim 2, Let $\ell^*$ denote the number satisfying $2^{\ell^*} \cdot \mu \leq \lambda^{(k)} < 2^{\ell^*+1} \cdot \mu$, by this choice we know $\mu/\lambda^{(k)} \geq \frac{1}{2}\exp(-3\ell^\star)$ we have

$$
\begin{aligned}
v_T^{(k)} \leq v_{B+\ell^*\hat{T}}^{(k)} &\leq v_B^{(k)}\exp(-3\ell^*) + 2\hat{T}\eta_{\ell^*}^2\lambda^{(k)}\sigma^2 \\
&\leq \frac{v_0^{(k)}}{T^3} + \frac{24\sigma^2}{\lambda^{(k)}T} + \frac{50\log_2\kappa}{3\lambda^{(k)}T} \cdot \sigma^2. \\
&\leq \frac{v_0^{(k)}}{T^3} + \frac{50\log_2\kappa}{\lambda^{(k)}T} \cdot \sigma^2.
\end{aligned}
$$

Therefore, the function value is bounded by $\mathbb{E}\left[f(\mathbf{w}_T^{\text{var}})\right] - f(\mathbf{w}^*) = \sum_{i=1}^d \lambda^{(k)}v_T^{(k)} \leq \frac{50\log_2\kappa}{T} \cdot \sigma^2 d$. $\qquad\square$

*Proof of proposition 3.* The proof of the proposition works similar to the proof of the strongly convex case of theorem 2, wherein, we combine the result of proposition 14 with lemma 9 and lemma 5 to obtain the result. $\qquad\square$

## D    Proofs of results in Section 3.3

All of our counter-examples in this section are going to be the same simple function. Let the inputs $x$ be such that only a single co-ordinate be active on each example. We refer to this case as the "discrete" case. Furthermore, let each co-ordinate be a Gaussian with mean $0$ and variance for the first $d/2$

directions being $d\kappa/3$ and the final $d/2$ directions being 1. Furthermore, consider the noise to be additive (and independent of $\mathbf{x}$) with mean zero. This indicates that $R^2 = \kappa$ for this problem.

Intuitively, we will show that in order to have a small error in the first eigendirection (with eigenvalue $\kappa$), one need to set a small learning rate $\eta_t$ which would be too small to achieve a small error in the second eigendirection (with eigenvalue 1). As a useful tool, we will decompose the variance in the two directions corresponding to $\kappa$ eigenvalue and 1 eigenvalue respectively as follows:

$$v_T^{(1)} \stackrel{\text{def}}{=} \mathbb{E}\left[\left(\mathbf{w}_T^{(1)} - (\mathbf{w}^*)^{(1)}\right)^2\right] = \prod_{j=1}^{T}(1 - \eta_j\kappa)^2 v_0^{(1)} + \kappa\sigma^2 \sum_{j=1}^{T}\eta_j^2 \prod_{i=j+1}^{T}(1 - \eta_i\kappa)^2$$

$$\geq \exp\left(-2\sum_{j=1}^{T}\eta_j\kappa\right)v_0^{(1)} + \kappa\sigma^2\sum_{j=1}^{T}\eta_j^2\exp\left(-2\sum_{i=j+1}^{T}\eta_i\kappa\right) \text{ and} \tag{15}$$

$$v_T^{(2)} \stackrel{\text{def}}{=} \mathbb{E}\left[\left(\mathbf{w}_T^{(2)} - (\mathbf{w}^*)^{(2)}\right)^2\right] = \prod_{j=1}^{T}(1 - \eta_j)^2 v_0^{(2)} + \sigma^2 \sum_{j=1}^{T}\eta_j^2 \prod_{i=j+1}^{T}(1 - \eta_i)^2$$

$$\geq \exp\left(-2\sum_{j=1}^{T}\eta_j\right)v_0^{(2)} + \sigma^2\sum_{j=1}^{T}\eta_j^2\exp\left(-2\sum_{i=j+1}^{T}\eta_i\right). \tag{16}$$

**Theorem 15.** *Consider the additive noise oracle setting, where, we have access to stochastic gradients satisfying:*

$$\widehat{\nabla f}(\mathbf{w}) = \nabla f(\mathbf{w}) + \zeta = \mathbf{H}(\mathbf{w} - \mathbf{w}^*) + \zeta,$$

*where,*

$$\mathbb{E}\left[\zeta | \mathbf{w}\right] = 0, \text{ and, } \mathbb{E}\left[\zeta\zeta^\top | \mathbf{w}\right] = \sigma^2\mathbf{H}$$

*There exists a universal constant $C > 0$, and a problem instance, such that for SGD algorithm with any $\eta_t \leq 1/2\kappa$ for all $t$[5], we have*

$$\limsup_{T \to \infty} \frac{\mathbb{E}\left[f(\mathbf{w}_T)\right] - f(\mathbf{w}^*)}{(\sigma^2 d/T)} \geq C\frac{\kappa}{\log(\kappa + 1)}.$$

*Proof.* Fix $\tau = \kappa/C\log(\kappa + 1)$ where $C$ is a universal constant that we choose later. We need to exhibit that the $\limsup$ is larger than $\tau$. For simplicity we will also round $\kappa$ up to the nearest integer.

Let $T$ be a given number. Our goal is to exhibit a $\tilde{T} > T$ such that $\frac{f(\mathbf{w}_{\tilde{T}}) - f(\mathbf{w}^*)}{(\sigma^2/\tilde{T})} \geq \tau$. Given the step size sequence $\eta_t$, consider the sequence of numbers $T_0 = T, T_1, \cdots, T_\kappa$ such that $T_i$ is the first number that

$$\frac{1}{\kappa} \leq \sum_{t=T_{i-1}+1}^{T_i}\eta_t \leq \frac{3}{\kappa}.$$

Note that such a number always exists because all the step sizes are at most $2/\kappa$. We will also let $\Delta_i$ be $T_i - T_{i-1}$. Firstly, from (15) and (16), we see that $\sum_t \eta_t = \infty$. Otherwise, the bias will never decay to zero. If $f(\mathbf{w}_{T_{i-1}+\Delta_i}) - f(\mathbf{w}^*) > \frac{\tau\sigma^2 d}{T_{i-1}+\Delta_i}$ for some $i = 1, \cdots, \kappa$, we are done. If not, we obtain the following relations:

$$\frac{\sigma^2}{\Delta_1} \leq \sigma^2\sum_{t=1}^{\Delta_1}\eta_{T_0+t}^2 \leq \frac{\exp(3)}{\kappa}\cdot\mathbb{E}\left[\left(\mathbf{w}_{T_0+\Delta_1}^{(1)} - (\mathbf{w}^*)^{(1)}\right)^2\right]$$

$$\leq \exp(3)(f(\mathbf{w}_{T_0+\Delta_1}) - f(\mathbf{w}^*)) \leq \frac{\exp(3)\tau\sigma^2}{T_0 + \Delta_1}$$

$$\Rightarrow T_0 \leq (\exp(3)\tau - 1)\Delta_1.$$

Here the second inequality is based on (15). We will use $C_1$ to denote $\exp(3)$. Similarly, we have

$$\frac{\sigma^2}{\Delta_2} \le \sigma^2 \sum_{t=1}^{\Delta_2} \eta_{T_1+t}^2 \le \frac{C_1}{\kappa} \mathbb{E}\left[\left(\mathbf{w}_{T_1+\Delta_2}^{(1)} - (\mathbf{w}^*)^{(1)}\right)^2\right] \le C_1(f(\mathbf{w}_{T_1+\Delta_2}) - f(\mathbf{w}^*)) \le \frac{C_1\tau\sigma^2}{T_1+\Delta_2}$$

$$\Rightarrow T_1 \le (C_1\tau - 1)\Delta_2 \quad \Rightarrow \quad T_0 \le \frac{(C_1\tau-1)^2}{C_1\tau}\Delta_2.$$

Repeating this argument, we can show that

$$T = T_0 \le \frac{(C_1\tau-1)^i}{(C_1\tau)^{i-1}}\Delta_i \quad \text{and} \quad T_i \le \frac{(C_1\tau-1)^{j-i}}{(C_1\tau)^{j-i-1}}\Delta_j \quad \forall\, i < j.$$

We will use $i = 1$ in particular, which specializes to

$$T_1 \le \frac{(C_1\tau-1)^{j-1}}{(C_1\tau)^{j-2}}\Delta_j \quad \forall\, j \ge 2.$$

Using the above inequality, we can lower bound the sum of $\Delta_j$ as

$$\sum_{j=2}^{\kappa}\Delta_j \ge T_1 \cdot \sum_{j=2}^{\kappa} \frac{(C_1\tau)^{j-2}}{(C_1\tau-1)^{j-1}} \ge T_1 \cdot \frac{1}{C_1\tau} \cdot \sum_{j=2}^{\kappa}\left(1 + \frac{1}{C_1\tau}\right)^{j-2}$$

$$\ge T_1 \cdot \frac{1}{C_1\tau} \cdot \exp\left(\kappa/(C_1\tau)\right). \tag{17}$$

This means that

$$\mathbb{E}\left[f(\mathbf{w}_{T_i})\right] - f(\mathbf{w}^*) \ge \frac{d}{2} \cdot \mathbb{E}\left[\left(\mathbf{w}_{T_i}^{(2)} - (\mathbf{w}^*)^{(2)}\right)^2\right] \ge \exp(-6)\sigma^2 d \cdot \sum_{i=1}^{\Delta_1}\eta_{T+i}^2$$

$$\ge \frac{\exp(-6)\sigma^2 d}{\Delta_1} \ge \frac{\exp(-6)\sigma^2 d}{T_1} \ge \frac{\exp\left(\kappa/(C_1\tau)-3\right)}{C_1\tau} \cdot \frac{\sigma^2 d}{\sum_{j=2}^{\kappa}\Delta_j},$$

where we used (17) in the last step. Rearranging, we obtain

$$\frac{\mathbb{E}\left[f(\mathbf{w}_{T_\kappa})\right] - f(\mathbf{w}^*)}{(\sigma^2 d/T_\kappa)} \ge \frac{\exp\left(\kappa/(C_1\tau)-3\right)}{C_1\tau}.$$

If we choose a large enough $C$ (e.g., $3C_1$), the right hand side is at least $\frac{\exp((C/C_1)\log(\kappa+1)-3)}{\kappa} \ge \kappa$. $\qquad \square$

*Proof of theorem 4.* Theorem 4 follows as a straightforward consequence of Theorem 15 and lemma 11. $\qquad \square$

**Theorem 16.** *There exists universal constants $C_1, C_2 > 0$ such that for any $\tau \le \frac{\kappa}{CC_1\log(\kappa+1)}$ where $C$ is the constant in Theorem 4, for any SGD algorithm and any number of iteration $T > 0$ there exists a $T' \ge T$ such that for any $\tilde{T} \in [T', (1+1/C_2\tau)T']$ we have $\frac{\mathbb{E}[f(\mathbf{w}_{\tilde{T}})]-f(\mathbf{w}^*)}{(\sigma^2 d/\tilde{T})} \ge \tau$.*

**Theorem 17.** *Consider the additive noise oracle setting, where, we have access to stochastic gradients satisfying:*

$$\widehat{\nabla f}(\mathbf{w}) = \nabla f(\mathbf{w}) + \zeta = \mathbf{H}(\mathbf{w} - \mathbf{w}^*) + \zeta,$$

*where,*

$$\mathbb{E}\left[\zeta|\mathbf{w}\right] = 0, \text{ and, } \mathbb{E}\left[\zeta\zeta^\top|\mathbf{w}\right] = \sigma^2\mathbf{H}$$

*There exists universal constants $C_1, C_2 > 0$ such that for any $\tau \le \frac{\kappa}{CC_1\log(\kappa+1)}$ where $C$ is the constant in Theorem 4, for any SGD algorithm and any number of iteration $T > 0$ there exists a $T' \ge T$ such that for any $\tilde{T} \in [T', (1+1/C_2\tau)T']$ we have $\frac{\mathbb{E}[f(\mathbf{w}_{\tilde{T}})]-f(\mathbf{w}^*)}{(\sigma^2 d/\tilde{T})} \ge \tau$.*

To prove Theorem 17, we rely on the following key lemma, which says if a query point $\mathbf{w}_T$ is bad (in the sense that it has expected value more than $10\tau\sigma^2 d/T$), then it takes at least $\Omega(T/\tau)$ steps to bring the error back down.

**Lemma 18.** *There exists universal constants $C_1, C_2 > 0$ such that for any $\tau \leq \frac{\kappa}{CC_1 \log(\kappa+1)}$ where $C$ is the constant in Theorem 4, suppose at step $T$, the query point $\mathbf{w}_T$ satisfies $f(\mathbf{w}_T) - f(\mathbf{w}^*) \geq C_1\tau\sigma^2 d/T$, then for all $\tilde{T} \in [T, (1 + \frac{1}{C_2\tau})T]$ we have $\mathbb{E}\left[f(\mathbf{w}_{\tilde{T}})\right] - f(\mathbf{w}^*) \geq \tau\sigma^2 d/T \geq \tau\sigma^2 d/\tilde{T}$.*

*Proof of Lemma 18.* Since $f(\mathbf{w}_T) - f(\mathbf{w}^*) \geq C_1\tau\sigma^2 d/T$ and

$$f(\mathbf{w}_T) \quad = \quad \frac{d}{2}\left(\kappa\left(\mathbf{w}_T^{(1)} - (\mathbf{w}^*)^{(1)}\right)^2 + \left(\mathbf{w}_T^{(2)} - (\mathbf{w}^*)^{(2)}\right)^2\right), \quad \text{we} \quad \text{know} \quad \text{either}$$

$\left(\mathbf{w}_T^{(1)} - (\mathbf{w}^*)^{(1)}\right)^2 \geq C_1\tau\sigma^2/2\kappa T$ or $\left(\mathbf{w}_T^{(2)} - (\mathbf{w}^*)^{(2)}\right)^2 \geq C_1\tau\sigma^2/2T$. Either way, we have a coordinate $i$ with eigenvalue $\lambda_i$ ($\kappa$ or 1) such that $\left(\mathbf{w}_T^{(i)} - (\mathbf{w}^*)^{(i)}\right)^2 \geq C_1\tau\sigma^2/(2T\lambda_i)$.

Similar as before, choose $\Delta$ to be the first point such that

$$\eta_{T+1} + \eta_{T+2} + \cdots + \eta_{T+\Delta} \in [1/\lambda_i, 3/\lambda_i].$$

First, by (15) or (16), we know for any $T \leq \tilde{T} \leq T + \Delta$, $\mathbb{E}\left[\left(\mathbf{w}_{\tilde{T}}^{(i)} - (\mathbf{w}^*)^{(i)}\right)^2\right] \geq \exp(-6)C_1\tau\sigma^2/(2\lambda_i T)$ just by the first term. When we choose $C_1$ to be large enough the contribution to function value by this direction alone is larger than $\tau\sigma^2/T$. Therefore every query in $[T, T + \Delta]$ is still bad.

We will consider two cases based on the value of $S^2 := \sum_{\tilde{T}=T+1}^{T+\Delta} \eta_{\tilde{T}}^2$.

If $S^2 \leq C_2\tau/(\lambda_i^2 T)$ (where $C_2$ is a large enough universal constant chosen later), then by Cauchy-Schwartz we know

$$S^2 \cdot \Delta \geq \left(\sum_{\tilde{T}=T+1}^{T+\Delta} \eta_{\tilde{T}}\right)^2 \geq 1/\lambda_i^2.$$

Therefore $\Delta \geq T/C_2\tau$, and we are done.

If $S^2 > C_2\tau/(\lambda_i^2 T)$, by Equation (15) and (16) we know

$$\mathbb{E}\left[\left(\mathbf{w}_{T+\Delta}^{(i)} - (\mathbf{w}^*)^{(i)}\right)^2\right] \geq \sigma^2 \sum_{\tilde{T}=T+1}^{T+\Delta} \eta_{\tilde{T}}^2 \exp\left(-2\lambda_i \sum_{j=\tilde{T}+1}^{T+\Delta} \eta_j\right)$$

$$\geq \exp(-6)\sigma^2 \sum_{\tilde{T}=T+1}^{T+\Delta} \eta_{\tilde{T}}^2 \geq \exp(-6) \cdot C_2\tau\sigma^2/(\lambda_i^2 T).$$

Here the first inequality just uses the second term in Equation (15) or (16), the second inequality is because $\sum_{j=\tilde{T}+1}^{T+\Delta} \eta_j \leq \sum_{j=T+1}^{T+\Delta} \eta_j \leq 3/\lambda_i$ and the last inequality is just based on the value of $S^2$. In this case as we can see as long as $C_2$ is large enough, $T + \Delta$ is also a point with $\mathbb{E}\left[f(\mathbf{w}_{T+\Delta})\right] - f(\mathbf{w}^*) \geq \lambda_i \mathbb{E}\left[\left(\mathbf{w}_{T+\Delta}^{(i)} - (\mathbf{w}^*)^{(i)}\right)^2\right] \geq C_1\tau\sigma^2/(T + \Delta)$, so we can repeat the argument there. Eventually we either stop because we hit case 1: $S^2 \leq C_2\tau/\lambda_i^2 T$ or the case 2 $S^2 > C_2\tau/\lambda_i^2 T$ happened more than $T/C_2\tau$ times. In either case we know for any $\tilde{T} \in [T, (1 + 1/C_2)T]$ $\mathbb{E}\left[f(\mathbf{w}_{\tilde{T}})\right] - f(\mathbf{w}^*) \geq \tau\sigma^2/T \geq \tau\sigma^2/\tilde{T}$ as the lemma claimed. $\square$

*Proof of Theorem 17.* Theorem 17 is an immediate corollary of Theorem 15 and Lemma 18. $\square$

*Proof of Theorem 16.* Theorem 16 is an immediate corollary of Theorem 17 and lemma 11 $\square$

# E    Details of experimental setup

## E.1    Synthetic 2-d Streaming Least Squares Experiments

As mentioned in the main paper, we consider four condition numbers namely $\kappa \in \{50, 100, 200, 400\}$. We run all experiments for a total of $\kappa_{\max}^2 = 400^2 = 160000$ iterations. The two eigenvalues of the Hessian are $1$ and $1/\kappa$ respectively and the noise level $\sigma^2 = 1$ and we average our results with five random seeds. All our grid search results for the polynomially decaying learning rates are conducted on a $8 \times 8$ grid of learning rates $\times$ decay factor and whenever a best run lands at the edge of the grid, the grid is extended so that we have the best run in the interior of the grid search. For the step decay schedules, note that we fix the learning rate (details below), and vary only the decay factor.

For the $O(1/t)$ learning rate, we search for decay parameter over $8-$points log-spaced between $\{1/(200\kappa), 5000/\kappa\}$. The starting learning rate is searched over $8$ points logarithmically spaced between $\{1/\kappa, 5\}$.

For the $O(1/\sqrt{t})$ learning rate, the decay parameter is searched over $8$ logarithmically spaced points between $\{1/(2500\kappa), 100/\kappa\}$. The starting learning rate is searched between $\{1/(10\kappa), 5\}$ with $8$ logarithmically spaced points.

For the step decay schedule experiments, we kept the initial learning rate to be $0.1$ and swept over when to decay in multiples of $T/\log T$, i.e., vary some parameter $c \in \{0.25, 0.5, 0.75, 1.0, 1.25, 1.5, 2, 4\}$ where the learning rate decays by a factor of $2$ every $c \cdot T/logT$ steps. We found that the values chosen in most experiments were very close to $1$, i.e., they were either $1$ or $1.25$ or some very rare cases, $1.5$.

With regards to the suffix iterate averaging, we used a constant stepsize of $0.1$ and averaged iterates over the final half of the iterations.

## E.2    Non-Convex experiments on cifar-10  dataset with a 44-layer residual net

As mentioned in the main paper, for all the experiments, we use the Nesterov's Accelerated gradient method [Nes83] implemented in pytorch [6] with a momentum set to $0.9$ and batchsize set to $128$, total number of training epochs set to $100$, $\ell_2$ regularization set to $0.0005$.

With regards to learning rates, we consider $10-$values geometrically spaced as $\{1, 0.6, \cdots, 0.01\}$. To set the decay factor for any of the schemes such as 5,6, and 7, we use the following rule. Suppose we have a desired learning rate that we wish to use towards the end of the optimization (say, something that is $100$ times lower than the starting learning rate, which is a reasonable estimate of what is typically employed in practice), this can be used to obtain a decay factor for the corresponding decay scheme. In our case, we found it advantageous to use an additively spaced grid for the learning rate $\gamma_t$, i.e., one which is searched over a range $\{0.0001, 0.0002, \cdots, 0.0009, 0.001, \cdots, 0.009\}$ at the $80^{th}$ epoch, and cap off the minimum possible learning rate to be used to be $0.0001$ to ensure that there is progress made by the optimization routine. For any of the experiments that yield the best performing gridsearch parameter that falls at the edge of the grid, we extend the grid to ensure that the finally chosen hyperparameter lies in the interior of the grid. All our gridsearches are run such that we separate a tenth of the training dataset as a validation set and train on the remaining $9/10^{th}$ dataset. Once the best grid search parameter is chosen, we train on the entire training dataset and evaluate on the test dataset and present the result of the final model (instead of choosing the best possible model found during the course of optimization).