[Reviews · NeurIPS 2019]

Reviewer 1



Originality: I'm not an expert on this subfield, but as far as I know the work is original. Quality: I think this is a very nice paper. The results are interesting and clearly stated. And the work addresses an important question about the difference between theoretical learning rate schedules / averaging schemes and those used in practice. Also the work touches on the important difference between general function classes (e.g. convex) and specific interesting instances (e.g. least squares). I have a couple of major comments: 1) organisation of the paper. In my opinion the CIFAR-10 experiments should be brought forward in the paper to follow the introduction. This is because as it stands it looks like the experiments are being run to test the theory. But I believe everybody in the community already knew these experimental results and conceptually they actually motivated the development of the theory. It should be made clear in the discussion that these CIFAR-10 experiments are not an honest attempt to falsify the theory. 2) that brings me to the second point that I believe more effort could be made to experimentally test the theory. For example, I believe the authors should run the Polyak averaging scheme on CIFAR-10 with polynomially decaying step sizes. If the reason that step-decay works better than poly-decay for final iterate is related to the theoretical arguments in the paper, then we would also expect Polyak averaging to work well as it has a similar theoretical foundation to the authors' algorithm. If it doesn't then that raises more interesting questions that could be highlighted and left for future work. Clarity: the work is very clear. Significance: I believe the work to be significant. I am giving score of weak reject to give the authors an impetus to address my points, but I expect to upgrade the score provided they either argue their side or agree to change the paper. I have not checked the math but intend to do so before the next round of review. Minor comments: 1. Figure 2, the caption says the plot has a log scale, but it doesn't. 2. An inconsistent notation is used for dot product. E.g. equation 1, equation after line 147, equation 2, equation 3. 3. Please could the authors write the assumptions in latex assumption blocks so they are easily visible at a glance.

Reviewer 2



*************After author response************ Thank you for the answer. I'll keep my mark and vote for accepting this paper. **************************************************** Originality and significance: as far as I understand the paper, the tools used to analyze the last iterate of SGD are not new, yet the results on least square are novel and deserve some attention. Note that there are only a few comments on the difference of behavior of the last iterate estimator for general convex problems and least-square problems in the case of polynomially decreasing step sizes. Quality and clarity : the main quality of the paper is its clarity and its accuracy. All the insights claims are sufficiently argued and developed during the theorems and the purpose of the paper is clear.

Reviewer 3



The paper studies the SDG algorithm for least square regression problems. A new analysis with step decay schedule of the learning rate is presented which theoretically shows the near optimality of the algorithm. The paper also proves that the polynomial decay schemes learning rate is sub-optimal. Overall, the paper is well-written and theoretically sound. I have not checked the proof in detail but the analysis looks right to me. Base on the theoretical novelty of the paper, I would recommend for publication.

[Author Response · NeurIPS 2019]

Thanks to the reviewers for their effort towards providing feedback on our paper.

**Reviewers 1 and 2 – iterate averaging for ResNet on CIFAR-10:** We will be happy to perform experiments evaluating the performance of iterate averaging for CIFAR-10 with a residual network and incorporate these results in the final version. These experiments are somewhat subtle since, performing Polyak averaging right from the beginning might not be good due to the non convex nature of the problem. One needs to find the right number of iterations after which we need to begin Polyak averaging. These experiments (on the performance of iterate averaging) could be of great help to the community since iterate averaging is frequently discussed but no thorough experiments results have been published to the best of our knowledge.

We will now address specific reviewer comments.

**Reviewer 1:** Thank you for your comments.

Moving CIFAR-10 experiments to the start of the paper: The suggestion on presenting the experiments section in the introduction is quite interesting. We will think about it carefully before the final version. Even if the CIFAR-10 experimental results are not strictly novel, our primary goal was to perform a thorough grid search based experiment in order to understand what is the behavior of the final iterate with both polynomial and exponentially decaying step size schedules. To the best of our knowledge, we are unaware of experiments that perform these experiments thoroughly and with reasonably large neural networks.

Minor comments : will address 1,2,3. Note that for 3, the x-axis is in log scale (instead of y-axis).

**Reviewer 2:** Thank you for your comments.

Experiments on least squares objective: We will present results on the least squares objective with both iterate averaging as well as polynomially and exponentially decaying step sizes. We also note that in figure 1 (right) of the paper, we present results with grid searches on optimizing a quadratic objective - see caption of figure 1 as well as appendix E.1 in the supplementary section. The (bad) performance of polynomially decaying learning schemes, as well as the (good) performance of exponentially decaying step sizes for the quadratic objective implies that these perform similarly for the least squares objective - this is through the result of lemmas 8-11 in the appendix.

What problems do these lower bounds hold for? Our construction relies on objective functions that have bad condition numbers, and that is rather typical for many machine learning problems. Furthermore, note that going beyond least squares, for objectives that satisfy notions of local quadratic approximation (for e.g. self-concordance), our results (after going through some more formal arguments) has the potential to be made to apply towards the rate near the optimum. We will include this discussion in the final version.

**Reviewer 3:** Thank you for your review.

[Meta-Review · NeurIPS 2019]

The paper considers SGD for least squares regression, and establishes results for the last iterate (as is often done in practice) as opposed to an average over many iterates (as is often in theory). Well written. Tools are not new, and so somewhat incremental in that sense, but the paper is well written and on a core problem, so is of interest in that sense.